

# Dissolved organic matter characteristics of deciduous and coniferous forests with variable management: different at the source, aligned in the soil

Lisa Thieme[1,6], Daniel Graeber[2], Diana Hofmann[3], Sebastian Bischoff[4], Martin T. Schwarz[5], Bernhard
Steffen[3], Ulf-Niklas Meyer[4,8], Martin Kaupenjohann[6], Wolfgang Wilcke[7], Beate Michalzik[4], Jan
Siemens[1]

[1] Institute of Soil Science and Soil Conservation, iFZ Research Centre for Biosystems, Land Use and Nutrition, Justus Liebig University Giessen, Heinrich-Buff-Ring 26-32, 35392 Giessen, Germany
[2] Department of Aquatic Ecosystem Analysis, Helmholtz Centre for Environmental Research - UFZ, Magdeburg, Brückstraße 3a, 39114 Magdeburg, Germany
[3] Institute of Bio- and Geosciences, Agrosphere (IBG-3), Forschungszentrum Jülich, Wilhelm-Johnen-Straße, 52428 Jülich, Germany
[4] Institute of Geography, Department of Soil Science, Friedrich Schiller Universität Jena, Löbdergraben 32, 07743 Jena, Germany.
[5] current address: Office of Landscape, Agriculture and Environment, Building Department, Canton of Zurich, Walcheplatz 2, 8090 Zurich, Switzerland
[6] Department of Ecology, Chair of Soil Science, Technische Universität Berlin, Ernst-Reuter-Platz 1, 10587 Berlin, Germany
[7] Institute of Geography and Geoecology, Karlsruhe Institute of Technology (KIT), Reinhard-Baumeister-Platz 1, 76131 Karlsruhe, Germany
[8] current address: Institute of Landscape Ecology, University of Münster, Heisenbergstrasse 2, 48141 Münster, Germany

*Correspondence to*: Lisa Thieme (l.thieme@campus.tu-berlin.de)





**Abstract.** Dissolved organic matter (DOM) is part of the biogeochemical cycles of carbon and nutrients, carries pollutants and drives soil formation. The DOM concentration and properties along the water flow path through forest ecosystems depend on its origin and transformation processes. To improve our understanding of the effects of forest management, especially tree species selection and management intensity, on DOM concentrations and properties of samples from different

ecosystem fluxes, we studied throughfall, stemflow, litter leachate and mineral soil solution at 26 forest sites in the three regions of the German Biodiversity Exploratories. We covered forest stands with three management categories (coniferous and deciduous age-class, unmanaged beech forests). In water samples from these forests, we monitored DOC concentrations over four years and characterized the quality of DOM with UV-vis absorption, fluorescence spectroscopy combined with parallel factor analysis (PARAFAC) and with Fourier-Transform Ion Cyclotron Resonance Mass Spectrometry (FT-ICR-

MS). Additionally, we performed incubation-based biodegradation assays. Multivariate statistics revealed strong significant effects of origin of ecosystem fluxes and smaller effects of main tree species on DOM quality. Coniferous forests differed from deciduous forests by showing larger DOC concentrations, more lignin- and protein-like molecules, and less tannin-like molecules in throughfall, stemflow, and litter leachate. Cluster analysis of FT-ICR-MS data indicated that DOM compositions, which varied in aboveground samples depending on tree species, become aligned in mineral soil. This

alignment of DOM composition along the water flow path in mineral soil is likely caused by microbial production and consumption of DOM in combination with its interaction with the solid phase, producing a characteristic pattern of organic compounds in forest mineral soils. We found similarly pronounced effects of ecosystem fluxes on the biodegradability of DOM, but surprisingly no differences between deciduous and coniferous forests. Forest management intensity, mainly determined by biomass extraction, contribution of species, which are not site-adapted, and deadwood mass, did not influence

DOC concentrations, DOM composition and properties.

**Introduction**

Dissolved organic matter (DOM) processing and transport is highly dynamic in forest ecosystems (Kaiser and Kalbitz, 2012) and plays an important role in the biogeochemical cycles of carbon and nutrients (Bolan et al., 2011; Kaiser and Kalbitz, 2012). The chemical composition of DOM strongly affects its role in the carbon and nutrient cycles of ecosystems (Bolan et

al., 2011). The chemical composition, in turn, depends on the DOM source and its processing along the water flow path through ecosystems.

Following the water path through a forest ecosystem, there are numerous sources and sinks of DOM: Rain water moves through the atmosphere, washes through forest canopy and understory vegetation, infiltrates and percolates the forest litter layer and the organic-matter-rich topsoil and passes further downward through the deeper mineral soil reaching groundwater

tables and entering the aquifer. Precipitation incorporates atmospheric aerosol ingredients like dust and gases, containing organic carbon (Aitkenhead-Peterson et al., 2003). Typical dissolved organic carbon (DOC) concentrations of precipitation range from 0.6 to 7.6 mg L-1 in Europe (Aitkenhead-Peterson et al., 2003). The below-canopy fluxes consist of throughfall

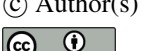



and stemflow both containing DOM of different quality (Moore et al., 2003; Inamdar et al., 2012; Levia et al., 2012; Levia and Germer, 2015; Michalzik et al., 2016). Organic compounds are released from leaves (Wickland et al., 2007), twigs and tree-stems (Levia and Germer, 2015), but also from insects (Michalzik et al., 2016) and bacteria (Lindow and Brandl, 2003, Müller et al., 2006) inhabiting the canopy and leaf surfaces. Important sources of DOM, especially at the soil surface, are

decomposition products of leaf litter (Cleveland et al., 2004; Klotzbücher et al., 2013) and deadwood or coarse woody debris (Kahl et al., 2012; Bantle et al., 2014; Magnusson et al., 2016). Major belowground sources of DOM are root exudates (Yano et al., 2000; Baetz and Martinoia, 2014, Tückmantel et al., 2017), microbial primary and secondary metabolites (Aitkenhead-Peterson et al., 2003), and degradation products of soil organic matter (DOM as left-over of soil organic matter degradation, e.g. Gödde et al., 1996; Hagedorn et al 2004).

The ecosystem fluxes (throughfall, stemflow, litter leachate and soil solution) in turn are influenced by forest management practices. Thus, the source of DOM is affected by changing the tree composition through partial or complete removal and/or replacement of specific tree species, by exporting biomass and by modifying the proportion of deadwood (Goldmann et al., 2015; Augusto et al., 2002). Various studies under laboratory and field conditions showed differences in litter leachate DOC concentrations, DOM biodegradability and compositions for different tree species (Cleveland et al., 2004; Don and Kalbitz,

2005; Klotzbücher et al., 2013; Cuss and Guéguen 2013). In the mineral soil, the chemical composition of root exudates appears to be species-specific and hence the microbial rhizosphere community associated with each plant species is different (Van Dam and Bouwmeester, 2016). Changing the amount and species of deadwood influences fungal community composition, wood decomposition and release of DOM quantity and quality (Arnstadt et al., 2016).

Both, sources and processing, affect DOM chemical composition (Stubbins et al., 2017). Biological DOM production and

mineralization are important mechanisms regulating DOM dynamics in the environment (Benner, 2002; Bolan et al., 2011). Biodegradability of DOM is, beside other controls, driven by intrinsic characteristics like molecular structure, functional group content or size of the molecules (Marschner und Kalbitz, 2003). During DOM transformation and mineralization by microorganisms, several classes of chemical compounds are preferentially oxidized to $CO_2$ (e.g. carbohydrates), while others passively accumulate as leftover, e.g. lignin, lipids and waxes (Kalbitz et al., 2003). Similarly, some fractions of DOM

are sorbed more strongly by components of the solid soil (minerals, organic matter) systematically changing the DOM quality (e.g. Kaiser et al., 1996).

We hypothesized that i) the composition and ii) the biodegradability of DOM changes systematically along the water flow path through forest ecosystem, from throughfall (TF), stemflow (SF), litter layer leachate (LL) to mineral topsoil (TOP) and subsoil (SUB) solution, whereby the DOM composition as well as the direction and magnitude of its changes depend on

main tree species and forest management intensity, measured as the Forest Management Index (ForMI, Kahl and Bauhus, 2014). To test these hypotheses, we assessed the quality, structural composition and bioavailability of DOM and its concentration measured as DOC in 26 differently managed forests in three regions in Germany. We characterized DOM quality using a combination of indices derived from UV-vis absorbance, fluorescence components derived from parallel factor analysis (PARAFAC) of fluorescence-excitation-emission-matrices (EEMs), and molecular formulae obtained with





high-resolution Fourier Transform Ion Cyclotron Resonance Mass Spectrometry (FT-ICR-MS). Additionally, we assessed DOM biodegradability in an incubation experiment.

## Material and methods

### Study sites

We conducted the study on experimental plots at the Schwäbische Alb (Alb), the Hainich-Dün (Hainich) and the Schorfheide-Chorin (Schorfheide) sites of the German "Biodiversity Exploratories", which were established as a platform for large-scale and long-term functional biodiversity research (DFG Schwerpunktprogramm 1374, www.biodiversity-exploratories.de). For sample collection, we selected nine forest plots in each of the Alb (AEW1-AEW9) and Hainich (HEW1-HEW6 and HEW10-HEW12) sites and eight forest plots in the Schorfheide sites (SEW1-SEW3 and SEW5-SEW9).

The forests comprise three management categories: i) unmanaged beech-dominated forests (Fagus sylvatica L., for at least 60 years), ii) beech-dominated (deciduous) age-class forests, and iii) coniferous age-class forests (spruce-dominated, Picea abies L. for Alb and Hainich and pine-dominated, Pinus sylvestris L. for Schorfheide). As a measure for forest management intensity, we used the forest management intensity indicator (ForMI) proposed by Kahl and Bauhus (2014). The ForMI is the sum of three management-related factors: the ratio of the harvested to total tree volume, the contribution of not site-adapted

tree species, and the contribution of deadwood volume with saw-cuts to the total deadwood mass. Higher ForMI values indicate a higher intensity of forest management. Important climatic and geological information of the three sites are given in Fischer et al. (2010). A summary as well as essential property of the investigated forest plots are given in the supporting information (Table S1).

### Sampling

We collected solution samples with a bi-weekly 2-day sampling routine from above- and below-ground ecosystem fluxes during the vegetation periods from April 2011 to November 2015. We sampled throughfall (TF), stemflow (SF), litter leachate (LL), mineral topsoil (TOP) and subsoil solutions (SUB) as volume-weighted composite samples of multiple individual samplers for each ecosystem flux.

We sampled TF with 20 funnel-type collectors (diameter 0.12 m, polyethylene, PE) per forest ecosystem, which we placed

0.3 m above the soil surface, arranged in two lines of 10 samplers in a cross-shaped form. To minimize alterations of the samples, e.g. by evaporation, photochemical reactions, or growth of algae, we wrapped the sampling bottles with aluminium foil and covered the opening of the collection bottle with a 1.6 mm polyester mesh and a table-tennis ball. We sampled SF with sliced polyurethane hoses (diameter: 0.04 m) fixed around tree stems and sealed with a polyurethane-based glue to the bark of three trees per site at approximately 1.5 m height. The polyurethane hose was connected with a polypropylene or

high density (HD) PE barrel via a PE tube. We collected forest floor litter leachate with three zero-tension lysimeters per site (280 cm2 sampling area each) consisting of polyvinyl chloride plates covered with a PE net (mesh width 0.5 mm) connected





via PE hoses to 2 L HDPE bottles (Nalgene®) stored in a box below-ground. We sampled soil solution with nylon membrane (0.45 µm) and borosilicate glass suction cups (ecoTech, Germany). Three suction cups were installed beneath the A horizon (TOP) at approximately 10 cm depth. Another three suction cups were installed in the B horizon (SUB) in approximately 50 cm depth. Because of shallow soils, this was not possible in the Alb plots. Suction cups were connected to 2-L HD-PE bottles (Nalgene®) in an insulated aluminium box placed into a soil pit. We extracted soil water by applying a vacuum of 50 kPa to the HDPE bottles with an electric pump after each sampling.

After recording sample volumes with graded cylinders and merging samples from individual samplers to volume-weighted composite samples per flux and plot in the field, we transported the samples on ice to the laboratory and stored them overnight at 5°C. In the laboratory, all samples were filtered through cellulose filters (Sartorius, Germany, Grade: 292) on the following day. We washed the filters with 100 mL deionized water and 10 mL of sample prior to filtration of the remaining sample and froze all filtered samples at -18°C until further analysis. Preliminary tests showed that freezing the samples decreased the measured DOC concentration by 5 % on average and also affected DOM fluorescence (Thieme et al., 2016). However, since the samples of all ecosystem fluxes (TF, SF, LL, TOP, SUB) were affected in a similar magnitude, freezing did not hamper the comparison of samples regarding changes in DOM quality and DOC concentration.

**Sample processing for optical and chemical characterization of DOM**

We thawed the samples over night at 8°C and conducted fluorescence and UV-absorption measurements without further preparations. We analyzed in total 466 Hainich and Schorfheide samples of all ecosystem fluxes and management categories taken between 2011 and 2013. To balance uneven sample numbers, we calculated mean EEMs per plot and ecosystem flux resulting in a dataset with 79 samples. For FT-ICR-MS analysis, we chose TF, SF, LL and SUB samples from unmanaged beech and coniferous age-class forests of the SCH sites in April and May 2015. To gain enough sample volume for the analysis, we pooled samples from two forest plots per management category gaining a total of 8 samples. After re-filtration (0.45 µm, Whatman GF/C), samples were desalted and concentrated using solid phase extraction (SPE, C18 Hydra cartridges, Machery & Nagel, Düren, Germany) using methanol (≥99.98 %, Ultra LC-MS grade; Carl Roth, Karlsruhe, Germany) as eluent. After SPE, the solution was dried at room temperature. Before FT-ICR-MS measurements, the samples were re-dissolved in methanol.

**Sample processing to assess the biodegradability of DOM**

We used TF, SF and LL samples from plots of all management categories collected in October 2012 to assess the biodegradability of DOM. In this study, we refer to biodegradable dissolved organic carbon (BDOC) as the DOC utilized by heterotrophic microbes via complete mineralization of C sources to obtain energy, and by incorporation of carbon into microbial biomass. For each management category and ecosystem flux, we pooled samples from two to three forests per site gaining a total number of 25 samples. We filtered the samples through a 0.2 µm Vacuflo filter in a laminar flow box beside a Bunsen burner and transferred 40 mL of the filtrate to sterile 250-mL suspension culture flasks (Greiner Bio-One,





Frickenhausen/Germany). After adding 2 mL of bacterial inoculum, we closed the flasks with semi-permeable caps. We incubated each sample in triplicate for seven time intervals (0, 3, 6, 10, 14, 20 and 28 days) at 20°C in the dark. Following the incubation, we filtered the samples through sterile 60-mL Soft-Ject single use syringes (Henke-Sass, Wolf; Tuttlingen/Germany) equipped with nylon syringe filters with a pore size of 0.45 µm (Rotilabo, Carl Roth;

Karlsruhe/Germany). The filtered samples were stored frozen until the measurement of DOC concentrations and the UV-vis and fluorescence spectra.

We prepared the bacteria inoculum by collecting and merging soil samples from forests of each site. We combined sieved, field moist soil from the first 10 cm after removing the litter layer of all three sites with unfiltered TF solution of the same sites with a soil:solution ratio of 1:10, subsequently shook the mixture for 30 min and centrifuged it for 10 min (Heraeus

Megafuge 16, Thermo Scientific; Waltham/USA). We stored the supernatant at 8°C prior to incubation.

An overview of sampling time and sample composition per analysis is given in Table 1. Detailed information of selected plots per site, number of measured samples per ecosystem flux and composition of pooled samples for all measurements is provided as supporting information (Table S2).

**Measurement of DOC concentrations, UV-vis absorption and fluorescence spectra**

We measured DOC concentrations (routine limit of quantification: 3 mg L-1) as non-purgeable organic carbon (NPOC) on a Shimadzu TOC-VCPH Analyzer (Shimadzu, Duisburg, Germany). Absorption spectra of DOM were recorded for wavelengths ranging from 400 nm to 600 nm using a Lambda 20 UV-vis spectrometer (Perkin Elmer, USA) equipped with a 1-cm quartz cuvette. Measurements were baseline-corrected using ultra-pure water and all sample spectra were blank subtracted (ultra-pure water, EVOQUA, Warrendale, USA). Fluorescence excitation emission matrices (EEMs) were

recorded on a Hitachi F-4500 fluorescence spectrometer (Hitachi, Japan) directly after absorption measurement in the same cuvette. We used excitation wavelengths ranging from 240 nm to 450 nm (5 nm steps) and emission wavelengths ranging from 300 nm to 600 nm (2 nm steps) with a slit width of 5 nm and scan speed of 12000 nm/min. We corrected our EEMs according to the protocol of Murphy (2010) with the fdomcorrect function in the drEEM toolbox (Murphy, 2013) using Matlab (Matlab, 2015a). For the excitation and emission correction factors, we used the supplies provided by the

manufacturer. We measured ultra-pure water fluorescence spectra for blank correction and for converting EEMs to Raman units by normalizing them to the area under the Raman peak at 350 nm excitation wavelength. We diluted the samples with ultra-pure water to ensure an absorption < 0.3 at 254 nm (Ohno, 2002) and subsequently performed the inner-filter correction, again using the fdomcorrect function in the drEEM toolbox (Lackowitz, 2006).

Using the absorbance spectra, we calculated specific ultraviolet absorbance (SUVA$_{254}$) as the absorbance at 254 nm per m

pathlength of light, divided by the concentration of DOC in mg L$^{-1}$, reported in L mg$^{-1}$ m$^{-1}$. The SUVA$_{254}$ index reflects the bulk aromaticity of DOM (Weishaar, 2003).





**DOM characterization using FT-ICR-MS and UV absorption**

Ultra-high-resolution mass spectra were acquired using an ESI-LTQ-FT Ultra instrument (ThermoFisher Scientific, San Jose, CA, USA) equipped with a 7 T supra-conducting magnet (Oxford Instruments, Abingdon, UK). The mass spectrometer was used in negative mode, tuned daily and calibrated following a standard optimization procedure for almost all settings.

Hence, the settings of the ion optics typically varied slightly from day to day. Samples were analyzed within three days as pure methanol solution without any pH modification or water addition. Typical standard conditions were: spray voltage 2.9 kV, capillary voltage -50 V, tube lens -93 V. Best performances for our sample set were received when sheath, auxiliary and sweep gas were turned off. The transfer capillary temperature was set to 275°C. Samples were introduced into the ESI source with a syringe pump at a rate of 5 µL min$^{-1}$. Mass spectra in profile mode were recorded in full scan from 200 to

1000 Da, measured at resolving 400.000 at m/z 400 Da (for complete separation of CHONS- from 13C1CHOS in even numbered peaks). Each individual mass spectrum contained 50 transients. The automatic gain control target in the ICR cell was set to 5 x E5 (for nearly negligible interactions between the ions) to achieve deviations considerably below 1 ppm (supplier specification). Six spectra were averaged for improving the statistical robustness of the final spectra that were further processed. The mean deviation of the raw spectra was approximately 0.4 ppm at m/z 400 Da, therefore all files were

recalibrated before calculation (to prevent two possible assignments as CHO and CHOS2, respectively, for the same peak, which would lead to excluding this mass from further consideration and therefore loss of information). Prior to and between some analyses, blanks were measured. Molecular formulae were assigned using an in-house developed post-processing Scilab routine (Scilab Enterprises 2012). For quality control, all peaks of at least two randomly selected masses (odd and subsequent even numbered, respectively) were characterized by hand to control the exactness of measured and recalculated

peaks, respectively, and to set proper constraints in the calculation program (maximal number of C, H, O, N and S atoms, respectively).

**Analysis of fluorescence and FT-ICR-MS data**

To identify the underlying fluorescence components of the DOM, we used parallel factor analysis (PARAFAC) to mathematically decompose the trilinear data of the EEMs (Stedmon, 2003). All further preprocessing steps of EEMs, like

smoothing of Rayleigh and Raman scatter and normalization, as well as the PARAFAC analysis were conducted with the *drEEM* toolbox (Murphy, 2013) in Matlab (Matlab, 2015a). We chose a six component PARAFAC model (referred as C1 to C6), visually checked the randomness of residuals and the component spectral loadings, split-half validated the model and generated the best fit by random initialization. For comparison in statistical analyses, we used the relative percentage distribution of the six PARAFAC components (% of the sum of fluorescence of all PARAFAC components) %C1 to %C6

instead of C1 to C6. Identified PARAFAC components were described by comparison with published PARAFAC models, either manually or by using the OpenFluor database (Murphy et al., 2014).





To analyze FT-ICR-MS data, we used van Krevelen plots (van Krevelen, 1950) to visualize and characterize the assigned molecular formulae gained from the raw MS spectra. Therefore, the elemental ratios of oxygen to carbon (O/C) and hydrogen to carbon (H/C) for each formula of CHO compounds were plotted. Depending on the position in the van Krevelen diagram, all assigned formulae can roughly be grouped according to major classes of biopolymers found in natural organic

matter like tannin, lignin, lipids, proteins, amino sugars, and hydrocarbons. We used the classification according to Sleighter and Hatcher (2007), applying these assignments: lipids (H/C = 1.7–2.25, O/C = 0–0.22), proteins (H/C = 1.5–2.0, O/C = 0.2–0.5), amino sugars, (H/C = 1.5–1.75, O/C = 0.55–0.7), carbohydrates, (H/C = 1.5–2.0, O/C = 0.7–1.0), lignin (H/C = 0.75–1.5, O/C = 0.2–0.6), tannins (H/C = 0.5–1.25, O/C = 0.6–0.95) and condensed hydrocarbons (H/C = 0.2–0.75, O/C= 0–0.7). The number of formulae in each class were then summed and normalized by the total number of assignable formulae for all

functional groups to produce a relative abundance (as percent) for the six classes of biopolymers (Tfaily et al., 2015). Additionally, we conducted a cluster analysis with the standardized peak intensities of assigned formulae using Jaccard´s distances and Ward´s method in R (R core team, 2015; vegan package, Oksanen et al., 2017) according to Ide et al. (2017) and Stubbins et al. (2017).

With a correlation analysis (Spearman`s Rank Order Correlation, *stats* package in R, R Core Team and contributors

worldwide, 2018), we linked the modeled PARAFAC components with the biochemical information resulting from FT-ICR-MS measurements. Here, we used the relative abundances of PARAFAC components and the relative abundances of biopolymers extracted from van Krevelen plots.

**Effect of ecosystem flux, tree species and management on DOM composition and biodegradability: calculations and statistical analysis**

We used permutational multivariate analyses of variance (*vegan* package in R, Oksanen et al., 2017, Euclidean distances) to assess the effect of ecosystem flux (TF, SF, LL, TOP, SUB), main tree species (deciduous or coniferous), management intensity (ForMI) and their interactions on DOM composition (PARAFAC components, $SUVA_{254}$). DOC concentration values were not included to separately investigate effects of the drivers of DOM composition and DOC quantity. To visualize the PERMANOVA results, we conducted a PCA (*vegan* package in R, Oksanen et al. 2017) with the same DOM

composition variables. For the PCA, we scaled all DOM composition variables to reach unit standard deviation.

We used a type II ANOVA (*car* package in R, Fox and Weisberg, 2011) with interaction to test whether ecosystem flux, main tree species or management intensity affected DOC concentration (model Df = 19, residual Df = 59). Here, we log-transformed DOC concentrations to improve normal distribution and homoscedasticity of the residuals.

We conducted univariate pairwise tests to assess effects of ecosystem flux for each of the PARAFAC components,

separately for deciduous and coniferous forests. Moreover, we tested separately for management categories (deciduous age-class, beech unmanaged and coniferous age-class forests), if the ecosystem flux had an effect on DOC concentration and $SUVA_{254}$. Finally, for DOC concentration and $SUVA_{254}$, we assessed pairwise differences of main tree species and management category for each of the ecosystem fluxes. If normal distribution of the residuals was given, we used pairwise t-



tests with Holm-Bonferroni correction (*stats* package in R; R Core Team, 2016), otherwise, we applied Nemenyi-Damico-Wolfe-Dunn tests (Monte-Carlo test variant with 50000 iterations, *coin* package in R; Hothorn et al., 2006).

To describe the degradation kinetic of our DOM samples, we fitted a single exponential model. We quantified the rate of biodegradation by the mineralization constant (k) based on measurements of DOC concentrations measured during the entire

incubation period. Changes in DOM composition during degradation were assessed by projecting the six components of the PARAFAC model on the EEMs from samples measured before and after 28 days of incubation.

With a paired PERMANOVA (*vegan* package in R; Oksanen et al., 2017) we tested the effect of incubation, ecosystem fluxes (TF, SF, LL, TOP, SUB) and management category (deciduous age-class, beech unmanaged and coniferous age-class) and their interactions on DOC concentration, SUVA254 and %PARAFAC components of incubated samples. Subsequently,

we used Wilcoxon rank sum tests as paired test (stats package in R, R Core Team 2016) to evaluate the effect of incubation on DOC concentrations, $SUVA_{254}$ and %PARAFAC values. Finally, we used Spearman`s Rank Order correlation (*stats* package in R, R Core Team 2016) to assess the relationships between all variables (%BDOC, k, $SUVA_{254}$, %PARAFAC).

## Results

**Drivers of DOC concentrations and $SUVA_{254}$ in the solution samples from different ecosystem fluxes**

Mean DOC concentrations varied among water samples collected from different ecosystem fluxes and depending on main tree species (Figure 1). Following the water flow path, mean concentrations of solutions from unmanaged beech and deciduous age-class forests roughly increased from TF (9 mg $L^{-1}$ and 8 mg $L^{-1}$) via SF (18 mg $L^{-1}$ and 28 mg $L^{-1}$) to LL (31 mg $L^{-1}$ and 26 mg $L^{-1}$). They remained similar in TOP (24 mg $L^{-1}$ and 31 mg $L^{-1}$) and decreased to SUB (13 mg $L^{-1}$ and 12 mg $L^{-1}$) samples. Mean DOC concentration in coniferous age-class forests reached a maximum in SF (90 mg $L^{-1}$) and

decreased continuously via LL (55 mg $L^{-1}$) and TOP (30 mg $L^{-1}$) to SUB (13 mg $L^{-1}$). The ANOVA showed a significant effect of ecosystem fluxes and main tree species on DOC concentrations (ANOVA, p< 0.001), but management intensity (ForMI) had no significant effect. Comparing the DOC concentrations of all ecosystem fluxes within each management category (beech unmanaged, deciduous age-class and coniferous age-class), only few statistical significant differences were found (Figure 1). We found no differences in DOC concentrations of all ecosystem fluxes between the differently managed

beech forests. DOC concentrations in TF, SF and LL from beech forests were significantly smaller than those from coniferous forests.

Mean $SUVA_{254}$ values (indicative of the aromaticity of the DOM) were similar for all ecosystem fluxes except LL independent of management category. Mean values for TF, SF, TOP and SUB were 1,6–2,6 L $mg^{-1}$ $m^{-1}$ with coniferous SF rising up to 2.9 L $mg^{-1}$ $m^{-1}$. Significantly higher $SUVA_{254}$ values (p<0.05) for LL samples compared with all other sample

types equaled 3.5–3.7 L $mg^{-1}$ $m^{-1}$.



**FTICR-MS characterization of DOM composition**

The FT-ICR-MS spectra revealed differences in the distribution and abundance of organic molecules of varying mass and composition between ecosystem fluxes and management categories (Figures 2 and 3).

The numbers of assigned formulae in coniferous forest samples were similar for water samples of all different ecosystem fluxes, ranging between 8126 and 9522. In contrast, we found a slightly higher number of assigned formulae for LL (10112) and SUB (13447) samples from unmanaged beech forests compared to TF (9878) and SF (5435) samples.

Elemental formulae of CHO compounds for all samples plotted as van Krevelen diagrams revealed distinct differences for all above-ground ecosystem fluxes between coniferous (pine) and unmanaged beech forests (Figure 4). While van Krevelen plots for all pine forest samples exhibit a distinct share of formulae with a H/C ratio of 1.2–1.6 and a O/C ratio of 0.3–0.6, there was a lack of them in the above-ground beech forest samples. The space covered in the diagrams by DOM compositions of the different tree species, became increasingly aligned following the water path (Figure 4).

Depending on their position in the van Krevelen diagram, we assigned molecular formulae to seven major bio-molecular classes according to Sleighter and Hatcher (2007). Comparing their relative abundances between water samples collected from varying ecosystem fluxes and main tree species (Table 2), we found distinct differences for both. While lignin-like formulae were the dominant molecules in all ecosystem fluxes of coniferous forest DOM (50–66 %), we found almost balanced shares of lignin- and tannin-like molecules for TF (20–35 %) and SF (39 –40 %) of unmanaged beech forest DOM. The share of tannin-like molecules generally increased from TF via SF to LL samples and decreased again to SUB samples independent of main tree species. The share of tannin-like molecules reached up to 70 % in beech forests and only up to 27 % in pine forests. The other compounds like protein-like compounds, lipid-like compounds, amino sugar-type compounds, and carbohydrate-like compounds hardly contributed to the total molecular composition. Only condensed hydrocarbons had additional, noticeable shares of molecule composition for pine and beech TF samples (15 % and 36 %, Table 2).

Cluster analysis with the numbers of molecules assigned to major groups of biomolecules showed three distinct clusters. One included the subsoil solution samples of both, the pine and beech forests stands in the Schorfheide, the second all remaining solution samples from other ecosystem fluxes of pine forests and the third the same for beech forests (Figure 5).

**PARAFAC components - description and correlation with biochemical compounds**

Analyzing the fluorescence samples collected from 2010 to 2013 (see Table1), we validated a six-component PARAFAC model describing the variation of the fluorescence of DOM. The components were referred to as C1 to C6. Two fluorescence components (C1 and C6) had single excitation and emission maxima, whereas the other four components (C2 to C5) showed two local excitation maxima alongside one emission maximum. Component C1 was characterized by an excitation maximum <250 nm and an emission maximum at 436 nm. C2 showed two peaks of local excitation maxima at 265 nm and 375 nm, having an emission maximum at 480 nm. C3 exhibited two local excitation maxima, one at wavelengths <250 nm and the second at a wavelength of 315 nm, combined with an emission maximum at 404 nm. C4 showed two local excitation





maxima at wavelengths <250nm and at a wavelength of 325 nm, with an emission maximum at 446 nm. The fourth component with two local excitation maxima (<250 nm and 350 nm) was C5, which showed an emission maximum at a wavelength of 428 nm. The fluorescence of component C6 was characterized by an excitation maximum at 280 nm and an emission maximum at 334 nm. For detailed spectra of all PARAFAC components see the supporting information (Figure

S2).

We applied the previously validated six-component PARAFAC model to the fluorescence spectra of the DOM samples that were also characterized using FT-ICR-MS spectra (see Table1), to explore the molecular chemical background of the underlying fluorescence patterns. We found a significant positive correlation (Spearman´s rho, $p<0.05$) between the relative contribution of the fluorescence component C2 and the relative number of tannin-like molecules identified by mass

spectrometry. Significant negative correlations occurred between %C2 and the fraction of identified protein-like and amino sugar-like molecules (Table 3). The relative contribution of fluorescence component C3 to overall fluorescence significantly and positively correlated with the fraction of molecules assigned to the class of lignin-like biopolymers, while a significant negative correlation (Spearman´s rho, $p<0.05$) was observed with the fraction of tannin-like molecules (Table 3). The relative contribution of PARAFAC component C6 to overall fluorescence positively correlated with the fraction of protein-

like and amino sugar-like molecules (Table 3).

While the relevance of fluorescence components C2 and C4 for the overall fluorescence intensity increased with increasing DOC concentrations of the undiluted original samples, the contribution of fluorescence component C1 decreased with increasing DOC concentrations (Table 3).

**Drivers of the PARAFAC components**

Considering the lack of significant differences between unmanaged and age-class beech dominated forests for DOC concentrations and $SUVA_{245}$ values, we focused on comparing deciduous and coniferous forests. With a mean share of 32–39 %, component C1 dominated the overall fluorescence of DOM samples from both forests (Figure 6). Comparing water samples from different ecosystem fluxes for shares of %C1 in between those two forest categories, we only found significant differences for LL samples (Wilcoxon-test, $p<0.05$). The mean contribution of tannin-like components C2 ranged from 12–

23 % of total fluorescence and differed significantly between samples of aboveground ecosystem fluxes of deciduous and coniferous forests, with samples from the former showing a larger share of C2 to total fluorescence than samples from coniferous forests (Wilcoxon-test, $p<0.05$). In contrast, the mean contribution of lignin-like C3 to total fluorescence (13–22 %) was similar for both forest categories. Component C4 contributed between 4–25 % to total fluorescence and showed significant differences between forest categories only for water samples collected from belowground ecosystem fluxes.

Fluorescence component C5 ranged from 0–18 % and similar to protein-like component C6 (3–13 %) showed significant differences between deciduous and coniferous forests only for SF samples.

When comparing the distribution of single PARAFAC components between samples from different ecosystem fluxes along the water flow path within each management category, we found for %C1 smallest shares in TF samples increasing to



maximum shares in SF and LL samples and again slightly decreasing from TOP to SUB samples. A similar trend was observed for the contribution of the tannin-like component %C2, but reaching maximum shares in LL and TOP samples before decreasing again in SUB. The lignin-like fluorescence component %C3 showed an opposite trend to %C2, with smallest contributions to total fluorescence in LL samples, increasing again via TOP to reach its maximum contribution in SUB samples (Figure 6). We found a decreasing mean contribution of component %C4 from TF to SUB samples interrupted by a slight increase in SF samples. The reverse trend was found for the fluorescence component C5. The mean share of the protein-like component %C6 of total fluorescence was largest in TF samples. This share decreased along the flow path in LL and TOP to slightly increase SUB samples.

**Drivers of spectroscopic DOM composition (absorbance and fluorescence)**

To comprehensively assess the drivers of DOM composition we combined our absorbance and fluorescence spectroscopic results. We found a significant effect of the origin of samples from different ecosystem fluxes on DOM composition variables including $SUVA_{254}$ and %PARAFAC components (PERMANOVA, $p = 0.001$) explaining 67 % of sample variance. Further, a significant, albeit small effect ($R2=0.01$) was found for main tree species (PERMANOVA, $p = 0.04$). When investigating the individual ecosystem fluxes in detail, significant differences (Wilcoxon test, $p<0.05$) were found for samples from above ground ecosystem fluxes between coniferous and deciduous forest stands especially for %C2 (tannin-like), but not for %C3 (lignin-like) and %C4. Prominent differences disappeared when following the water underground, except for %C4 for which significant differences appeared. No significant effects were found for management intensity alone (PERMANOVA, $p =0.964$).

A PCA illustrated the distinctly different DOM composition in the water samples collected from various ecosystem fluxes (Figure 7). The first two components identified by the PCA explained 88% of the total variance (PC1: 60 %, PC2: 28 %). TF and SF were closely grouped together and differentiated from TOP and SUB along PC1, based most strongly on the different contributions of C4 and C5 to overall fluorescence. LL was separated especially from TF and SUB samples along PC2, based predominantly on their larger $SUVA_{254}$ and smaller contribution of C6 to overall fluorescence (Figure 7).

**DOM biodegradability**

We found a significant (Wilcoxon rank sum test, $p<0.001$) decrease of DOC concentrations with increasing time of incubation for all samples. The decrease could be adequately described using a single two parameter exponential model. Calculated degradation rate constants (k) were significantly different from zero for all samples and ranged between 0.004 d$^{-1}$ to 0.021 d$^{-1}$. SF proved to be the samples with the highest extent (Table S3) and rate of DOC degradation followed by TF with slightly lower values (Figure 8). In SF samples, 15–40% and in TF samples 17–35% of initial DOC was degraded within 28 days. With 8–18% of BDOC, LL samples showed two times lower values of degradation and up to 10 times lower rate constants than SF and TF. No significant differences for %BDOC and k were found between coniferous and deciduous forests.



SUVA$_{254}$ values showed a significant increase during the incubation (Wilcoxon rank sum test, $p<0.001$) for water samples from all ecosystem fluxes. The mean increase was lowest for TF (0.5 L mg$^{-1}$ m$^{-1}$) and similar for SF and LL (1.0 L mg$^{-1}$ m$^{-1}$). We found a significant negative correlation between %BDOC and SUVA$_{254}$ (Spearman´s rho, $p<0.05$).

We applied the previously validated six-component PARAFAC model on the EEMs of samples measured before and after 28 days of incubation. A significant increase after 28 days was found for %C1 for TF and SF samples (Wilcoxon rank sum test, $p<0.01$), but not for LL samples. We found a significant decrease of %C3 and %C4 during the incubation for TF samples only (Wilcoxon rank sum test, $p<0.01$). Although SUVA$_{254}$ was positively correlated with PARAFAC components %C1 and %C2, and negatively correlated with %C3 and %C6, no correlations were found between %BDOC and these PARAFAC components.

## Discussion

### Change of the DOM composition along the water flow path and among different forest management categories

Mean DOC concentrations of water passing through the forest ecosystems in our study followed concentrations reported in previous studies. For TF DOC concentrations documented in the literature ranged 2–35 mg L-1 (Michalzik et al., 2001; Moore, 2003; Stubbins et al., 2017), for SF 12–95 mg L$^{-1}$ (Moore, 2003; Levia et al., 2012; Stubbins et al., 2017) and for LL 14–90 mg L$^{-1}$ (Michalzik et al., 2001; Ide et al., 2017; Stubbins et al., 2017). Investigating soil solutions, others reported DOC concentrations 7–4 mg L$^{-1}$ for topsoil (Moore, 2003; Fellman et al., 2008b; Kindler et al., 2011; Ide et al., 2017) and 2–5 mg L$^{-1}$ for subsoil solutions (Michalzik et al., 2001; Peichl et al., 2007; Kindler et al., 2011). This pattern indicates that water is enriched in DOM during aboveground ecosystem passage and depleted while passing through mineral soil horizons. Also consistent with findings in other studies, SUVA$_{254}$ values of our DOM samples ranged 1.8- 4.7 L mg$^{-1}$ m$^{-1}$ for TF (Peichl et al., 2007; Inamdar et al., 2012; Stubbins et al., 2017), 1.9–11.2 L mg$^{-1}$ m$^{-1}$ for SF (Levia et al., 2012; Stubbins et al., 2017) and between 2.7–5.2 L mg$^{-1}$ m$^{-1}$ for LL (Peichl et al., 2007; Inamdar et al., 2012). This indicates an increasing share of aromatic DOM compounds when passing through the aboveground forest ecosystem. Reported ranges for topsoil solutions (2.2–3.9 L mg$^{-1}$ m$^{-1}$) and for subsoil samples (1.4–2.7 L mg-1 m$^{-1}$) are again similar to our findings (Peichl et al., 2007; Fellman et al., 2008b; Inamdar et al., 2012). The decrease in SUVA$_{254}$ values of DOM during the mineral soil passage could be related with preferential sorption of the aromatic DOM fractions (Kaiser and Guggenberger, 2000; Peichl et al., 2007).

Our ESI FT-ICR-MS measurements of forest DOM samples generated spectra with thousands of peaks, the amount and distribution of which were comparable with previous studies of natural DOM samples (e.g., Stenson et al., 2003; Sleighter et al., 2010; Tfaily et al., 2015). Due to the ultrahigh mass resolution, combined with the exactness in the sub ppm-range, it is possible to assign molecular formulae unique to almost all of the detected masses. The molecular composition of single peaks in our forest DOM samples led to their classification in typical biomolecular groups (lignin-like, tannin-like, condensed hydrocarbon-like, protein-like, amino sugar-like, lipid-like and carbohydrate-like), and was similar to those



reported by others for TF, SF, LL and subsoil solution samples (Tfaily et al., 2015; Ide et al., 2017; Stubbins et al., 2017). Consistent with other studies of DOM samples from aboveground ecosystems fluxes in oak and cedar forests (Stubbins et al., 2017), in wetlands (Hertkorn et al., 2016), and a fulvic acid isolated from small lake in Antarctica (D´Andrilli et al., 2013) , CHO-only compounds were the main fraction of assigned molecules.

Using the OpenFluor database (Murphy et al., 2014), we found close matches of component C1 with fluorescence components from studies in various environments characterized as "humic-like with terrestrial origin" (Santos et al., 2010; Yamashita et al., 2010b; Kothawala et al., 2012; Shutova et al., 2014; Dainard et al., 2015). Studies by Stedmon et al. (2003) in a Danish estuary and by Lambert et al. (2016) with Congo River water found components with spectra matching our component C2. They described this component also as "humic-like with terrestrial origin". The positive correlation with the

number of m/z peaks assigned to tannin-like compounds based on their position in the van Krevelen plots ($\rho$= 0.75, Table 3) along with the high contribution of C2 to the fluorescence found in LL and TOP (Figure 6) indicated that component C2 contained plant-derived, tannin-like components. Component C3 resembled components described as "microbially altered humic material" (Murphy et al., 2011), which were, among others, found in humic substances from sediments, in fen and bog pore water and in lakes, streams and estuaries (Santín et al., 2009; Shutova et al., 2014; Tfaily et al., 2015; Osburn et al.,

2016). C3 showed excitation and emission wavelengths ($\lambda_{ex}$= 250, 300nm; $\lambda_{em}$= 400nm) similar to those published in studies investigating fluorescence of lignin from different sources (e.g. Thruston, 1970; Albinsson et al., 1999). We found a significant positive correlation of the contribution of C3 to total fluorescence with the number of m/z peaks assigned to lignin-like compounds detected using FT-ICR-MS ($\rho$ = 0.80, Table 3). Therefore, we suggest that the share of C3 to total fluorescence reflected lignin and lignin-derived degradation products in DOM. As another "humic-like" component C4 was

termed "C peak" by Coble et al. (1996), which matched fluorescence components found by Kothawala et al. (2012) studying Swedish lakes as well as by studies investigating river and lake water (Lambert et al., 2016; Osburn et al., 2016). The humic-like component C5 only matched a component in the OpenFluor database that was reported by Lambert et al. (2016) studying Congo River water. The component C5 also falls into the EX/EM range of a component described as "humic-like C" by Coble et al. (2014) with sources referred also as "humic" and "terrestrial". The fluorescence of component C6 was

similar to the fluorescence of tryptophan and was therefore described as "protein-like", representing fluorescence of free amino acids and such bound in proteins. The component was included in numerous PARAFAC models of fluorescence of DOM from various environments (e.g. Murphy et al., 2013; Yu et al., 2015). The positive correlation between the protein-like as well as amino sugar-like fraction of FT-ICR-MS data and %C6 ($\rho$ = 0.74, Table 3) seems to confirm that protein-like fluorescence represented the fluorescence of proteins.

Our results showed distinct differences in DOC concentrations and DOM properties between water samples of forest ecosystem fluxes. TF was enriched in DOC (9–17 mg L$^{-1}$) relative to precipitation (2–5 mg L$^{-1}$) measured at the same sites during the same sampling period. In line with other studies (Peichl et al., 2007; Inamdar et al., 2012), low values for optical DOM properties like SUVA$_{254}$ and humic PARAFAC components C1 and C2 indicated a less "humic-like" and less aromatic DOM composition in TF compared with the other aboveground ecosystem fluxes. According to this interpretation,



we would expect low percentages of molecules assigned to the lignin, tannin and condensed hydrocarbons fractions gained by FT-ICR-MS analysis of TF samples. However, this was only found for tannin-like not for lignin-like compounds or condensed hydrocarbons (Table 2).

Elevated shares of condensed hydrocarbons in TF compared with the other ecosystem fluxes (Table 2) agreed with findings

of Stubbins et al. (2017) studying oak and cedar TF and SF samples. They suggested that atmospheric deposition of combustion products (primary aerosol) in combination with their reaction products (secondary aerosol) caused this noticeable fraction in DOM, due to accumulation from atmospheric aerosols on leaf surfaces. Combustion products have been shown to contribute to the designated condensed hydrocarbon fraction in the van Krevelen diagram (Kim et al., 2003; 2004).

TF samples were also richest in N-containing compounds, as shown by the highest relative contribution of component C6 (Figure 6) and of the protein-associated fraction of FT-ICR-MS molecules (Table 2) of all ecosystem fluxes. Beside free and bound proteins, atmospheric trace gases of bio- and anthropogenic origin react preferentially in the night with $NO_3$ radicals generating nitrogen organic compounds (Ervens et al., 2010; Farmer et al., 2010; Lee et al., 2016; Ng et al., 2017) possible to deposit in the (wet) canopy. This observation is also in line with findings in other studies (Inamdar et al., 2012; Ide et al.,

15  2017).

While following the water passage downward into subsoil layers, the decreasing DOC concentrations and $SUVA_{245}$ values, as well as decreasing percentages of tannin-like compounds (Table 2) were in line with a preferential sorption of aromatic, polyphenolic DOM in mineral soils (Kaiser und Guggenberger, 2000; Avneri-Katz et al., 2017). We had expected the fraction of molecules assigned to lignin in the van Krevelen plot to follow this behavior. Contrarily, we found increasing

concentrations of lignin-like compounds with increasing soil depth, which were less sorbed on the mineral phase than the highly-oxidized compounds associated with tannin-like molecules. Additional evidence of the on-going microbial processing of DOM along the flow path is the increasing share of the microbial-derived PARAFAC component C3 (Figure 6). The accumulation of lignin-like compounds may also be explained by a different interpretation of the van Krevelen diagram. It was suggested by others that the space covered by lignin molecules in the van Krevelen diagram should not only be linked to

higher plant source material, but also to other types of compounds proposed to be refractory, including non-aromatic compounds like carboxylic-rich alicyclic molecules (CRAM, Hertkorn et al., 2006; Stubbins et al., 2010; D'Andrilli et al., 2013).

We found only few significant differences between the distribution of PARAFAC components between different forest management categories prior to and after incubation. This could be attributable to balanced changes of the relative shares of

PARAFAC components used for comparison. Additionally, only parts of the DOM are able to absorb light potentially emitting light by fluorescence (Aiken, 2014). The combination of the low sensitivity of fluorescence and only small differences and/or changes in DOM composition during incubation might cause no visible changes of fluorescence.

Cluster analysis of biomolecules according to molecular composition (Figure 5) revealed the influence of tree species on aboveground DOM characteristics. Following the water downward, DOM properties assessed with FT-ICR-MS of



coniferous stands and deciduous forests from the same site converged, so that both SUB samples of both forest types grouped in one cluster. The same observation was true for all fluorescence components, except %C4, confirming that significant differences in properties detected with FT-ICR-MS disappeared between TOP and SUB samples.

We found significant differences in DOC concentrations of all aboveground ecosystem fluxes between deciduous and coniferous forests (Figure 1). Higher DOC concentrations in coniferous than beech forests might partly be attributable to differences in tree traits, like canopy and bark structure, and thus different water-vegetation contact times (Guggenberger et al., 1994).

The compositional differences between coniferous and deciduous aboveground DOM were mainly related to differences in the fractions associated with aromatic compounds like lignin and tannin (Table 2). We found a higher share of lignin-like compounds for water samples from pine compared to beech forests as revealed in the patterns of the van Krevelen plots (Figure 4), which agreed with findings of Ide et al. (2017). Additionally, we found different lignin-tannin ratios for both management categories. While pine samples exhibited up to 10-fold higher shares of lignin-like than tannin-like compounds, the ratio was close to or even smaller than one, especially in LL beech samples (Table 2). Tannins are secondary plant metabolites and play a role in herbivore defense and additionally may affect ecosystem processes (Kraus et al., 2003). A particularly large number of tannin-like molecules in solution samples from beech forests was also reflected in significantly higher shares of PARAFAC component C2 for TF, SF and LL samples in solution samples of beech than of pine (Figure 6). This agrees with findings of Lorenz et al. (2004), who reported higher concentrations of tannins in beech leaf litter than in pine needles. A higher share of phenolic carbon in beech than spruce solution samples from the same plots than this study was also found by Bischoff et al. (2015), based on 13C NMR analysis.

Besides the effect of different main tree species, we found no statistically significant effect of management practice on DOM composition. There were no difference between deciduous age-class and unmanaged beech forests as well as no influence of forest management intensity (ForMI) on optical DOM properties and DOC concentrations. With the ForMI, we applied an index that is only based on attributes related to aboveground vegetation (harvested tree volume, non-natural tree species, and deadwood volume with saw-cuts).

**Change of biodegradability along the water flow path and among different forest management categories**

The biodegradability of DOM in our solution samples was mainly determined by the type of ecosystem flux. The amount of %BDOC in all our samples of ecosystem fluxes in the range found by Qualls and Haines (1992) in deciduous forests samples (22–57 %). With a cumulative degradation of up to 40 % of the initial DOC concentration as well as the highest degradations rates, DOM from SF samples was most bioavailable. TF samples with BDOC up to 36 % contained DOM that seemed slightly less bioavailable. Lowest degradation rates and, thus, the most stable DOM were found in LL (8–18 %), comparable with results of Kalbitz et al. (2003), who reported mean values of 8% BDOC when incubating extracts from spruce and beech forest fermentation layers (Oa horizons).



Given the findings that carbohydrates and amino acids were typically preferentially utilized by microorganisms during degradation of different compounds in DOM solutions (Volk et al., 1997; Amon et al., 2001; Kalbitz et al., 2003), we expected a significant decrease of %C6 after 28 days of incubation. The fact that we found no significant change of %C6 during incubation might indicate that amino acids were bound in and on other, less degradable organic substances, so that

they were protected against degradation (Volk et al., 1997). However, phenolic compounds such as tannins and simple phenols have also been shown to contribute to those regions of fluorescence (Goldberg und Weiner, 1993; Maie et al., 2007; Hernes et al., 2009).

Consistent with other studies (Kalbitz et al., 2003; Fellman et al., 2008a), we found a negative correlation between %BDOC and aromaticity indicators (SUVA$_{254}$). This supported the assumption that especially aromatic structures are stable against

rapid biological degradation. The significant positive correlation between SUVA$_{254}$ and %C1 combined with the significant increase of component C1 after 28 days of incubation indicated either a transformation of former non-aromatic into aromatic compounds or a relative accumulation of the latter.

The larger share of condensed hydrocarbons in TF could explain the reduced biodegradability of DOM in TF than in SF. LL showed the highest portion of aromatic DOM compounds. This was indicated by the highest SUVA$_{254}$ values, the highest

percentage of the tannin-associated PARAFAC component C2 (Figure 6), and the highest share of the tannin-like and lignin-like molecules (Table 2). This observation coincided with studies of Peichl et al. (2007) and Inamdar et al. (2012) and could explain the lowest amount of %BDOC in LL compared to TF and SF.

**Conclusion**

There are distinct changes in DOC concentrations, chemical DOM composition, and DOM biodegradability along the water

flow path through European forested ecosystems. Aboveground DOM composition was influenced by forest management, namely selection of main tree species (deciduous versus coniferous), but not by management intensity (age-class beech versus unmanaged beech forests; ForMI). Biodegradability mainly depended on the type of ecosystem flux with SF containing the most biodegradable DOM and LL the least. The systematic changes of DOM properties suggest that the biotransformation and degradation of organic molecules in combination with their interaction with the soil solid phase cause

an alignment of the composition of DOM from different sources along the water flow path through forest ecosystems, producing a characteristic pattern of organic compounds in mineral soil solutions.

**Acknowledgements**

We thank the managers of the three Exploratories, Swen Renner, Kirsten Reichel-Jung, Sonja Gockel, Kerstin Wiesner, Katrin Lorenzen, Andreas Hemp, Martin Gorke, and all former managers for their work in maintaining the plot and project

infrastructure; Simone Pfeiffer, Maren Gleisberg, and Christiane Fischer for giving support through the central office, Jens





Nieschulze, and Michael Owonibi for managing the central data base, and Eduard Linsenmair, Dominik Hessenmöller, Daniel Prati, François Buscot, Ernst-Detlef Schulze, Wolfgang W.Weisser, and the late Elisabeth Kalko for their roles in setting up the Biodiversity Exploratories project. The work has been funded by the DFG Priority Program 1374 "Infrastructure Biodiversity Exploratories" (contributing project BECycles, Wi 1601/12, Ka 1139/17, Mi 927/2, Si 1106/4).

Field work permits were issued by the responsible state environmental offices of Baden-Württemberg, Thüringen, and Brandenburg (according to §72 BbgNatSchG).

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





## Tables

**Table 1: Overview of the samples used for the various analytical procedures. TF=Throughfall, SF=Stemflow, LL= Litter Leachate, SUB= Subsoil Solution**

| part of study | period of Sampling | sites | management category (number of investigated plots per site) | ecosystem fluxes | number of analized samples |
|---|---|---|---|---|---|
| **DOM characterization** | | | | | |
| *DOC + fluorescence* | April 2011 -November 2013 | Hainich, Schorfheide | unmanaged (3) deciduous age-class (2) coniferous age-class (3) | TF, SF, LL, TOP, SUB | 466 (79[b]) |
| *FTICR-MS* | April/Mai 2015 | Schorfheide | unmanaged (2) coniferous age-class (2) | TF, SF, LL, SUB | 8 [c] |
| **DOM biodegradability** | October 2012 | Alb, Hainich, Schorfheide | unmanaged (3) deciduous age-class (3/2)[a] coniferous age-class (3) | TF, SF, LL | 25 [d] |

[a] 3 plots for Alb site and 2 plots for Hainich and Schorfheide sites

[b] mean DOC and fluorescence spectra per plot and ecosystem flux used for all statistical analyses

[c] pooled solution samples per management category and ecosystem flux

[d] pooled solution samples per site, management category and ecosystem flux



**Table 2: Number of formulae (relative shares) assigned to major groups of biomolecules according to Sleighter and Hatcher (2007) obtained from FT-ICR mass spectra of DOM samples from ecosystem fluxes of coniferous (pine) and unmanaged beech forest from the Schorfheide. TF=Throughfall, SF=Stemflow, LL= Litter Leachate, SUB= Subsoil Solution**

| biomolecular groups | ecosystem flux | formulars within each ecosystem flux | |
|---|---|---|---|
| | | coniferous forest | unmanaged beech forest |
| *lignin-like* | TF | 840 (53%) | 194 (20%) |
| | SF | 1173 (50%) | 229 (39%) |
| | LL | 1088 (59%) | 108 (14%) |
| | SUB | 2735 (66%) | 2619 (63%) |
| *tannin-like* | TF | 96 (6%) | 345 (35%) |
| | SF | 309 (13%) | 231 (40%) |
| | LL | 503 (27%) | 583 (77%) |
| | SUB | 205 (5%) | 512 (12%) |
| *protein-like* | TF | 74 (5%) | 5 (1%) |
| | SF | 98 (4%) | 39 (7%) |
| | LL | 24 (1%) | 0 (0%) |
| | SUB | 67 (2%) | 48 (1%) |
| *amino sugar-like* | TF | 17 (1%) | 3 (0%) |
| | SF | 35 (2%) | 10 (2%) |
| | LL | 8 (0%) | 0 (0%) |
| | SUB | 27 (1%) | 16 (0%) |
| *lipid-like* | TF | 0 (0%) | 2 (0%) |
| | SF | 7 (0%) | 1 (0%) |
| | LL | 0 (0%) | 0 (0%) |
| | SUB | 5 (0%) | 3 (0%) |
| *cellulose-like* | TF | 1 (0%) | 1 (0%) |
| | SF | 21 (1%) | 2 (0%) |
| | LL | 4 (0%) | 0 (0%) |
| | SUB | 53 (1%) | 52 (1%) |
| *condensed hydrocarbons-like* | TF | 235 (15%) | 358 (36%) |
| | SF | 45 (2%) | 30 (5%) |
| | LL | 21 (1%) | 49 (6%) |
| | SUB | 89 (2%) | 145 (4%) |



**Table 3: Spearman´s rho for the correlation between the percentage relative abundances of PARAFAC components (%C1-%C6) and the relative abundances of biopolymers extracted from FT-ICR-MS van Krevelen plots. Significance level: * = p<0.05; ** = p< 0.01.**

|  | %C1 | %C2 | %C3 | %C4 | %C5 | %C6 |
|---|---|---|---|---|---|---|
| *DOC* | **-0.67\*** | **0.73\*** | -0,37 | **0.71\*** | -0,15 | -0,01 |
| *lipid-like* | 0,6 | -0,39 | 0,07 | 0,15 | -0,34 | 0,48 |
| *protein-like* | 0,43 | **-0.72\*** | 0,27 | 0,34 | -0,53 | **0.74\*** |
| *amino sugar-like* | 0,45 | **-0.64\*** | 0,21 | 0,42 | -0,56 | **0.74\*** |
| *lignin-like* | 0,05 | -0,5 | **0.80\*\*** | -0,41 | 0,31 | 0,18 |
| *tannin-like* | -0,29 | **0.72\*** | **-0.66\*** | 0,26 | 0,01 | -0,49 |


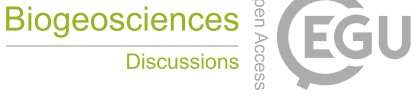

**Figures**

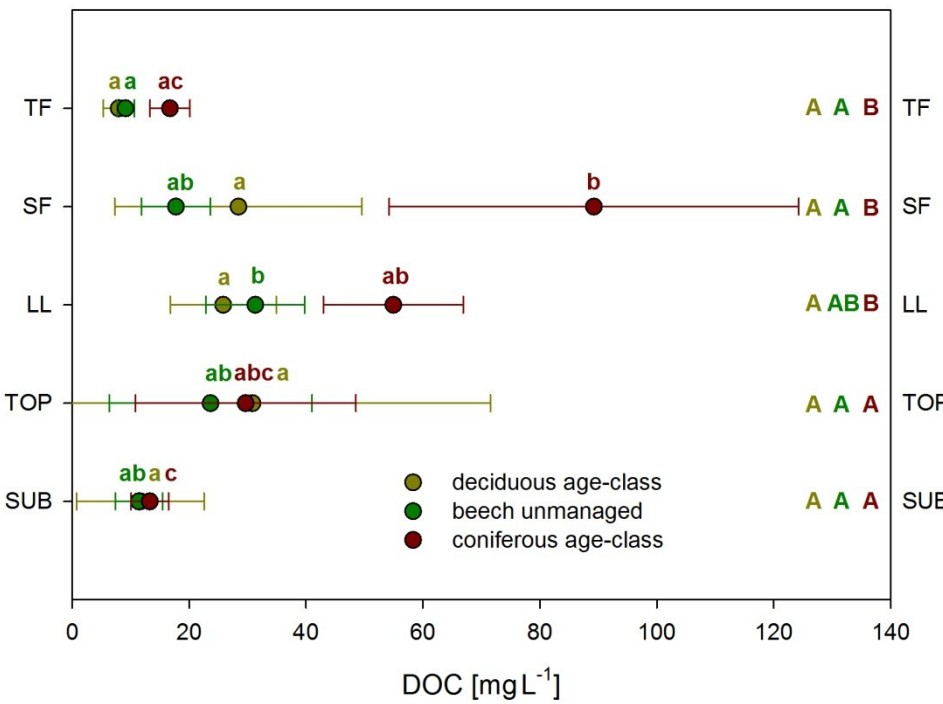

Figure 1: Mean DOC concentrations in ecosystem fluxes (TF, SF, LL, TOP, SUB) grouped according to management categories (deciduous age-class, beech unmanaged, coniferous age-class). Whiskers show standard deviations. TF=Throughfall n=201; 224; 244, SF=Stemflow n=140; 207; 140, LL= Litter Leachate n=179; 199; 203, TOP= Topsoil Solution n=60; 87; 47, SUB= Subsoil Solution n=63; 56; 65. Capital letters (reading horizontally): differences between management categories; lowercase (reading vertically): differences between water samples collected from different ecosystem fluxes within the same management category.



**Figure 2: Electrospray ionization Fourier transformation ion cyclotron mass spectra (ESI-FT-ICR-MS) of coniferous (pine) age-class forest samples from the Schorfheide (a-d) and detail for 499 m/z (e-h). Assigned molecular formulae in green and blue. TF=Throughfall, SF=Stemflow, LL= Litter Leachate, SUB= Subsoil Solution.**



**Figure 3: Raw electrospray ionization Fourier transformation ion cyclotron mass spectra (ESI FT-ICR-MS) of unmanaged beech forest samples from the Schorfheide (a-d) and detail for 499 m/z (e-h). Assigned molecular formulae in green and blue TF=Throughfall, SF=Stemflow, LL= Litter Leachate, SUB= Subsoil Solution**





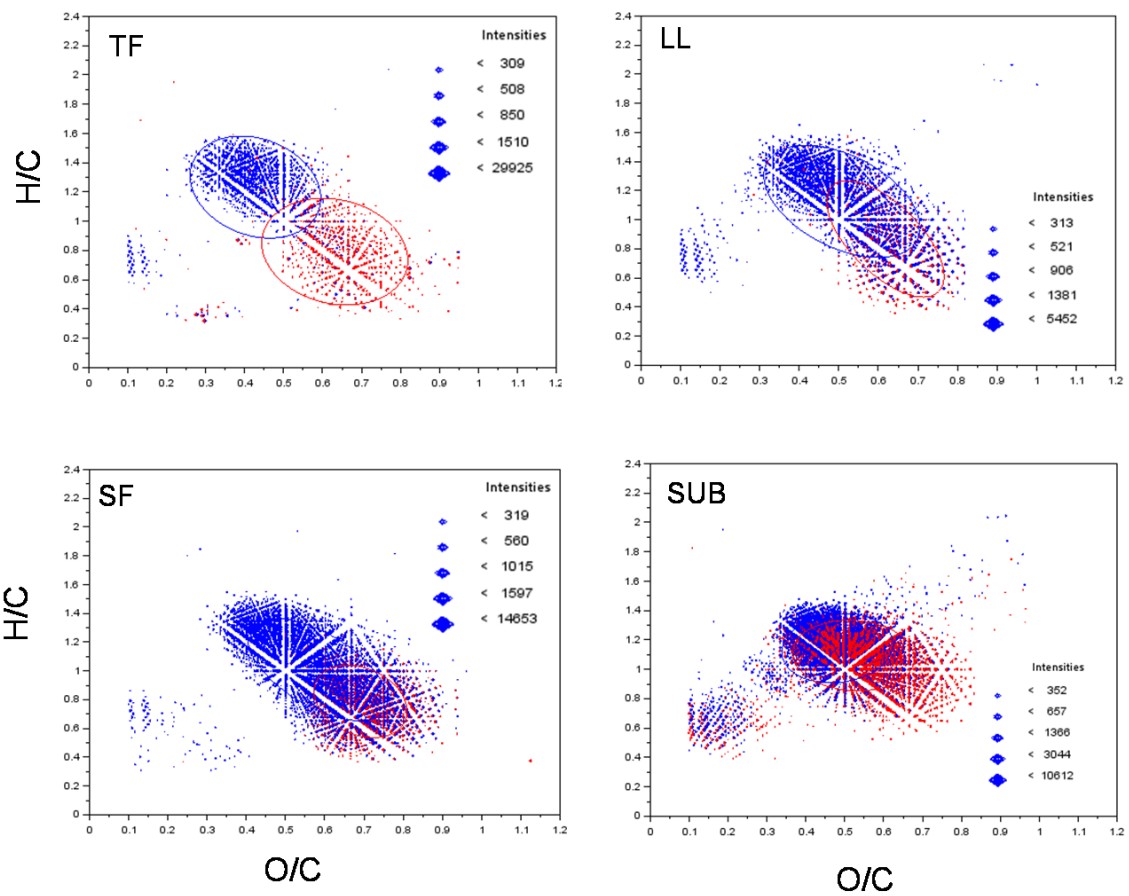

**Figure 4: Van Krevelen plots of CHO compounds for unmanaged beech (red) and coniferous (pine, blue) forest DOM samples. Ellipsoids indicate space covered by DOM samples. TF=Throughfall, SF=Stemflow, LL= Litter Leachate, SUB= Subsoil Solution**



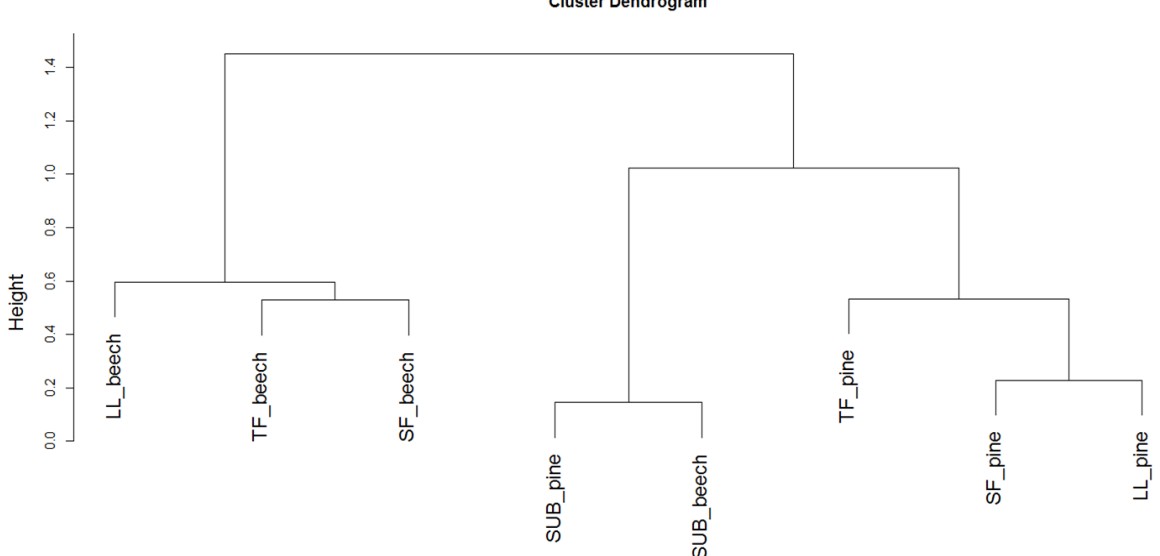

**Figure 5: Cluster dendrogram of number of molecules assigned to major groups of biomolecules (tannin-like, lignin-like, lipid-like, protein-like, amino sugar-like , and hydrocarbon-like ) according to Sleighter and Hatcher (2007) obtained from FTI-CR mass spectra of DOM samples from ecosystem fluxes of coniferous (pine) and unmanaged beech forest from Schorfheide sites.**
5    **TF=Throughfall, SF=Stemflow, LL= Litter Leachate, SUB= Subsoil Solution**





**Figure 6: Mean distribution of PARAFAC components in samples from different ecosystem fluxes of deciduous (age-class and unmanaged) and coniferous forests. Letters (reading vertically) indicate differences between samples from different ecosystem fluxes regarding PARAFAC components within each management category (Nemenyi-DamicoWolfe-Dunn test). TF = Throughfall, SF = Stemflow, LL = Litter Leachate, TOP = Topsoil Solution, SUB = Subsoil Solution**





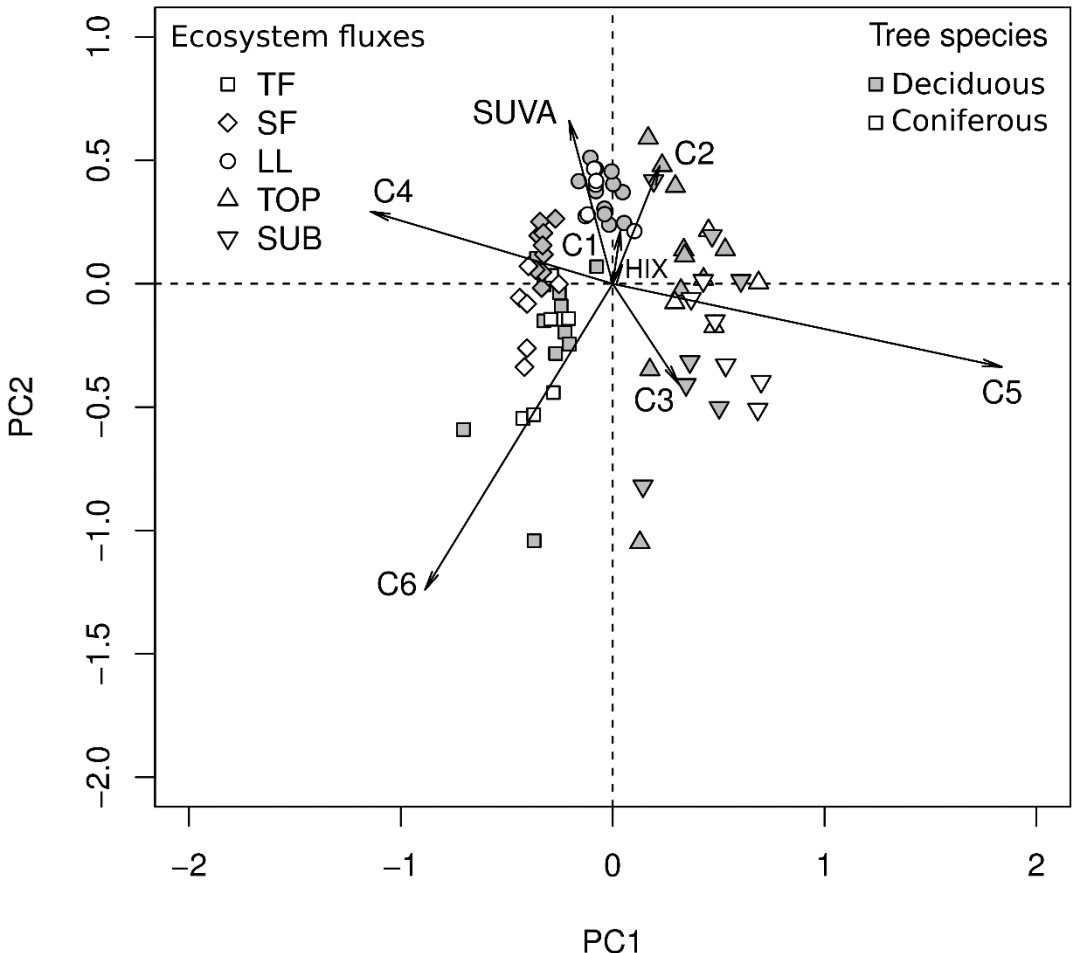

**Figure 7: PCA plot of DOM composition variables (SUVA$_{254}$, PARAFAC components C1-C6). TF = Throughfall, SF = Stemflow, LL = Litter Leachate, TOP = Topsoil Solution, SUB = Subsoil Solution. Variables n=7, samples n=79**





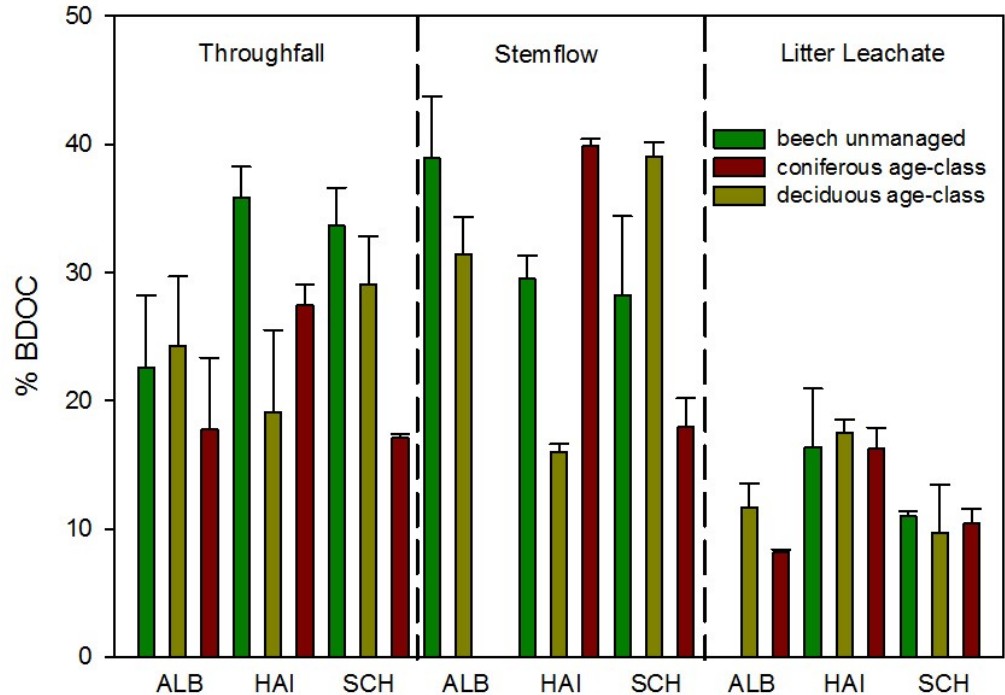

**Figure 8: Percentage biodegradable DOC (%BDOC) after 28 days of incubation in samples collected from different ecosystem fluxes. Bar chart: mean and SD of three replicates. TF = Throughfall, SF = Stemflow, LL = Litter Leachate**