# Peer review of "Dissolved organic matter characteristics of deciduous and coniferous forests with variable management: different at the source, aligned in the soil"

_Biogeosciences, 2018_

## Referee Comment (RC1) · Anonymous Referee #1 · 26 Nov 2018

The manuscript presents a study on quantities and qualities of DOM in decisive water fluxes (throughfall, stemflow, litter leachates and subsoil solutions) of temperate forests in dependence of tree species composition and forest management. The large data set available (26 individual sites, four years of sample collection) is mostly well evaluated, summarized and presented by the authors. The manuscript thus can make a significant contribution to research on biogeochemical nutrient cycling in such ecosystems. It contains a bulk of detailed information about how individual properties of DOM change along the water pathways, which provides insights into processes controlling the nutrient cycles. The conclusion that transformation processes in mineral soils cause an alignment of the properties of DOM from different sources is interesting and can help

to built novel conceptual models about the C cycle of terrestrial ecosystems.

Yet, one major shortcoming of the manuscript is that the properties of the soils of the study sites are not well described. There is some information about parent material and soil type given in the supplementary material, however, in my opinion a study on DOM in subsoil requires more detailed information about soil properties that may strongly affect the movement of DOM such as soil texture, mineral composition of the reactive clay fraction and pH (i.e., properties known to determine the chemistry of sorption processes). Hence, the focus of the discussion is a bit too much on biological factors of DOM movement, while the geo-chemical controls of soil processes should be covered a bit more. In case the data on soil mineralogy and chemistry are not available, I suggest to (at least) enhance the discussion on basis of available literature about how differences or similarities in geo-chemical factors between sites might have influenced the results of the present study. For instance, a decrease in highly oxidized compounds (page 15, line 21) might be explained by binding of the carboxyl-groups to positively charged surfaces of Al-/Fe-oxides at acidic pH values. In my view, the conclusion that the alignment of the composition of DOM is due to biotransformation and "interaction with the soil solid phase" needs support by considerations about such processes.

In addition to this major comment I only have a few more minor comments:

- Page 3, Line 27: The hypotheses could be stated more precisely, i.e., currently a broad predicition is made ("DOM changes systematically") without any consideration about the main mechanisms. How and why should the composition and biodegradability of DOM change along the water pathway? How and why should tree species and forest management affect these changes? - Page 3, Line 30: I suggest to briefly explain the ForMI here so that the readers can gain a better understanding of the study approach. - Page 4, Line 17: The work of Fischer et al. is not given in the reference list. - Results section: Although it is not the focus of the manuscript, it may be interesting to briefly summarize the magnitude of the temporal differences in DOC concentrations in the text (e.g., between the years and over the vegetation period); it is not clear to

me whether the temporal differences are mirrored in the standard deviations shown in Figure 1?

---

## Referee Comment (RC2) · Van Stan (Referee) · 3 Jan 2019

This manuscript presents a geographically and analytically broad study of dissolved organic matter (DOM) in forest types common throughout Europe. Little work has looked beyond DOC concentration and UV-vis absorption for all DOM fluxes along the rainfall-to-soil flow pathway in forests. In fact, this is one of only a handful of studies to include both throughfall and stemflow fluorescence, even fewer studies report biodegradable DOC proportions for these fluxes, and I'm aware of only 1 other study that has examined these fluxes using FTICR-MS. The large dataset appears to have been carefully examined and the methods are mostly well-described and appear well-done. Results

shed new insights on DOM processes along the canopy-to-soil flow pathway and appear to support the theory put forth in the title (different at the source, aligned in the soil). However, there are a few weaknesses that I believe should be addressed before publication:

Biodegradability measurement – were the samples spiked with nutrients to achieve N, P >Redfield limitations? All other details of the bioincubation tests look good. But, if we want to test the biolability of the DOC, then it is important to release the microbes from as many common limitations as possible (e.g. the authors set an optimal, controlled temperature: page 6, line 2). As a test of how much DOC is utilizable by microbes, this is a test of DOC quality, not an environmental rate at which one would expect the DOC to be utilized. Thus, ensuring the microbes are released from nutrient limitations, arguably, should be standard to allow comparison of DOC quality across studies, sites, between research groups, and independent of differences in C:N:P across environments. If this was not done, I recommend the authors briefly discuss the implications (biodegradation of DOC could have been constrained).

There are no measurements/estimates/tests of soil geochemical interactions with infiltrating DOM. As indicated above, biodegradation will likely be limited in natural settings (compared to the bioincubation tests – even for bioincubation tests without the nutrient spiking). I noticed that the other reviewer also believed this to be a shortcoming of the manuscript. As gathering more data along this vein would be difficult (and is, of course, not possible for storms already past unless it was collected at the time), I recommend the other reviewers' solution: provide more discussion of geochemical controls over DOM processes within soils. Perhaps the discussion could have subsections dedicated to biological factors and geochemical factors?

The meaning of DOM "origin" is unclear. For example, in the abstract: "strong significant effects of origin of ecosystem fluxes" – what is the "origin"? (A) Is it the first contact between precipitation and terrestrial surfaces (in the tree canopy), thus species-specific throughfall v. stemflow v. litter leachate? Or, (B) Is it the origin of specific DOM

fluorophores/molecular formulas? If (per A) the "origin" variable is used to indicate the initial DOM-enrichment process - throughfall or stemflow or litter leachate (for gap throughfall) - how is this different from the "species" variable? If the "origin" variable is used to indicate the origination of specific indicator fluorophores (like the component C1, "humic-like with terrestrial origin") or FTICR-MS formulas (like the N-rich organic compounds assumed to have atmospheric origins [p. 15, lines 10-15]), then this should be explicitly defined.

Although there is little literature covering throughfall and stemflow DOM quality, the authors missed some studies. Normally, one cannot cite all the studies on a particular topic; however, in this case, since so few studies exist, I recommend their inclusion. Please note that, for one of these papers, I am the lead author and it is not my intention to push my own work, only to account for the few studies on the topic. Introduction and discussion: Throughfall and stemflow DOM concentration, flux and quality (including potential sources and fates) have been reviewed and evaluated by Van Stan & Stubbins, 2018, https://doi.org/10.1002/lol2.10059. Page 16, lines 26-30: The authors only reference Qualls & Haines (1992) biodegradation estimates for throughfall. But, they do not discuss the only study reported stemflow BDOC in Qualls' recent Special Issue (Howard et al., 2018, https://doi.org/10.3390/f9050236).

---

## Referee Comment (RC3) · Tfaily (Referee) · 8 Jan 2019

This paper by Thieme et al. utilizes different analytical tools to characterize Dissolved organic matter characteristics of deciduous and coniferous forests with variable management. This is a very well written paper. The authors provided a lot for details regarding their sampling schemes, their approaches and their analysis. Very well done. I had few minor comments regarding the incubation experiments, geochemical data of the sites including pH, as well as FTICR MS data processing. The authors have generated a lot of data, however I felt the discussion was a bit weak or weaker than the rest of the paper. I would have loved to see more focus on both the biological and geochemical controls on soil processes rather than just explaining the data in context on how it compares to other studies. At certain times, the discussion felt as if it was a repetition of the results rather than generating hypothesis about the cause of such differences and providing some strong hypothesis regarding control on SOM degradation and the effect of management. Page 5, line 31, did you check if o.2 um was enough to eliminate microbial communities originally present in the samples? Pages 4-5 what was the total number of samples and how was it distributed in terms of management? Line 16, can you give the break down for the 466 samples? Line 16, what do you mean by: To balance uneven sample numbers, we calculated mean EEMs per plot and ecosystem flux resulting in a dataset with 79 samples. Did you collapse the 466 samples into 79 samples to allow for plot versus plot comparison? what was the variability within the same plot? Line 20, how did the optical data look for these samples? Do you believe that differences due to management is higher than that between plots within the same forest? Page 7, line 14, only six spectra were averaged? Typically, we do at least 100. Can you provide more details regarding formula assignment and the rules that were used? What was the number of unassigned formula? What were the ranges of C, H, N, C, OS, P? that were used? Figure 2, its hard to see the zoom in but be careful about peak splitting as this can affect your formula assignment. Even though it is hard to see, it appears you had some peak splitting.

---

## Author Comment (AC1) · 7 Feb 2019

Thank you for your detailed review of our manuscript. In the following, we hope to adequately address your constructive comments and questions.

"Yet, one major shortcoming of the manuscript is that the properties of the soils of the study sites are not well described. There is some information about parent material and soil type given in the supplementary material, however, in my opinion a study on DOM in subsoil requires more detailed information about soil properties that may strongly affect the movement of DOM such as soil texture, mineral composition of the reactive clay fraction and pH (i.e., properties known to determine the chemistry of sorption

processes). Hence, the focus of the discussion is a bit too much on biological factors of DOM movement, while the geo-chemical controls of soil processes should be covered a bit more. In case the data on soil mineralogy and chemistry are not available, I suggest to (at least) enhance the discussion on basis of available literature about how differences or similarities in geo-chemical factors between sites might have influenced the results of the present study. For instance, a decrease in highly oxidized compounds (page 15, line 21) might be explained by binding of the carboxyl-groups to positively charged surfaces of Al-/Fe-oxides at acidic pH values. In my view, the conclusion that the alignment of the composition of DOM is due to biotransformation and "interaction with the soil solid phase" needs support by considerations about such processes"

Being part of the DFG priority program 1374 offered the unique opportunity to access various kinds of information about the study sites. Therefore, it was possible to add additional geochemical information like soil texture and elemental composition. We will add the the new table S2 you can find in the supplement to this reply to the supporting information.

To assess whether geo-chemical processes controlled the DOM quality in our study, we additionally examined the relation between changes in DOC concentration and the ratio between organic carbon content of the mineral soil and the sum of its oxalate-extractable Fe- and Al-content (OC/[$Fe_o$+$Al_o$]) and compared the results with findings of Kindler et al. (2011).

We will ad Fig.1 of these reply as new Fig.2 to the manuscript We therefore will expand the discussion section as follows:

Page 13 Line 13: In the study of Kindler et al. (2011), the retention of DOC in mineral soil, expressed as percentage reduction of downward DOC flux, was closely related to the ratio between organic carbon concentration of the mineral soil and the sum of its oxalate-extractable Fe and Al content (OC/[$Fe_o$+$Al_o$]). This suggests that the DOC retention in mineral soils is governed by the sorption to the surfaces of Fe- and

Al-(hydr)oxides. Furthermore, organic matter sorption decreased exponentially with increasing organic matter coverage of the hydroxide surfaces (Kindler et al. 2011), which suggests that these hydroxides have a limited DOC sorption capacity (our hypothesis ii). In contrast to the findings of Kindler et al. (2011), we compare DOC concentrations, not fluxes. In order to test whether the data of our study fit the findings of Kindler et al. (2011), we plotted changes in DOC concentrations reported by Kindler et al. (2011) together with the data of this study against the ratio of OC/(Feo+Alo) in one graph (Figure 2). Different from fluxes, which always decreased with increasing soil depth in the Kindler et al. (2011) study, DOC concentrations increased with increasing depth at the Hainich sites with the highest OC/(Feo+Alo) ratios of all study regions (Figure 2). This increase in concentrations can be explained by a concentration effect because of evapotranspiration, if the DOC sorption capacity of pedogenic Fe- and Al-(hydr)oxides is saturated. Overall, the retention of DOC in the Hainich soils of this study fitted well to the DOC retention in the European data set of Kindler et al. (2011), who showed that the regional variation in DOC retention can be as large as the variation at continental scale. The concurrent decrease of DOC concentrations and increase of OC/( Feo+Alo) ratios between TOP and SUB (p = 0.027; Figure 2), corroborates the hypothesis that sorption to pedogenic Fe- and Al-(hydr)oxides controls DOC retention in mineral soils (Kindler et al. 2011). However, the results for mineral soils of the Schorfheide sites did not meet this pattern, as DOC concentrations decreased from TOP to SUB by 33-72% regardless of the OC/( Feo+Alo) ratio (Figure 2). At the Schorfheide sites, other processes than sorption to Fe- and Al-(hydr)oxide surfaces likely governed DOC retention. The Schorfheide soils developed from fluvioglacial quartzitic sands covering carbonate-free glacial till. Because of their poor pH buffering capacity, these soils were very acidic (pHCaCl2 = 3.0 – 3.6 in topsoils). The mean pH values in soil water samples of the Schorfheide sites was 4.5 in TOP solutions, increasing to 5.5 in SUB solutions. This means that Al-(hydr)oxides were dissolved in the Schorfheide topsoils, increasing Al3+-concentrations in soil water and leachates. The pH increase to 5.5 along the way from TOP to SUB likely induced a re-precipitation of Al. We assume that

dissolved organic matter transported from TOP to SUB co-precipitated together with Al3+ as described by Nierop et al. (2002) and Jansen et al. (2003, 2005) for acidic sandy soils from the Netherlands. If DOM was immobilized as insoluble metal-organic matter precipitate in B horizons, sorption sites on surfaces of pedogenic (hydr)oxides may be less important.

The following references will be added to the manuscript:

Jansen, B., Nierop, K.G.J., and Verstraten, J.M.: Mobility of Fe(II), Fe(III) and Al in acidic forest soils mediated by dissolved organic matter: influence of solution pH and metal/organic carbon ratios. Geoderma, 113, 323– 340, 2003.

Jansen, B., Nierop, K.G.J., and Verstraten, J.M.: Mechanisms controlling themobility of dissolved organic matter, aluminium and iron in podzol B horizons. Eur. J. Soil Sci., 56, 537–550. doi: 10.1111/j.1365-2389.2004.00686.x, 2005.

Nierop, K.G.J., Jansen, B., and Verstraten, J.M.: Dissolved organic matter, aluminium and iron interactions: precipitation induced by metal/carbon ratio, pH and competition. Sci. Total. Environ., 300, 201–211, 2002.

"Page 3, Line 27: The hypotheses could be stated more precisely, i.e., currently a broad predicition is made ("DOM changes systematically") without any consideration about the main mechanisms. How and why should the composition and biodegradability of DOM change along the water pathway? How and why should tree species and forest management affect these changes?"

To state our hypotheses more precisely we will change the manuscript as follows:

Page 3 Line 27: We hypothesized i) that the composition and the biodegradability of DOM changes to less polar, highly aromatic compounds with decreased bioavailability along the water flow path through forest ecosystems, from throughfall (TF), stemflow (SF), litter layer leachate (LL) to mineral topsoil (TOP) and subsoil (SUB) solution. We postulated ii) that aboveground changes of DOC concentrations and DOM composition

are mainly controlled by selective biological degradation, while changes in mineral soil are governed by sorption to mineral surfaces. Moreover, we hypothesized iii) that main tree species as well as forests management intensity affect the DOM composition as well as the direction and magnitude of its changes. The former because of the presence of species-specific compounds in DOM like phenolic secondary metabolites in beech forests, the latter, measured as the Forest Management Index (ForMI), beside others because of its influence on the biomass production and C input into the soil (Kahl and Bauhus, 2014).

"Page 3, Line 30: I suggest to briefly explain the ForMI here so that the readers can gain a better understanding of the study approach."

The ForMI is explained now in more detail on page 4, lines 15ff of the revised manuscript.

"Page 4, Line 17: The work of Fischer et al. is not given in the reference list."

Thank you for pointing this out. The reference will be added in the reference section

"Results section: Although it is not the focus of the manuscript, it may be interesting to briefly summarize the magnitude of the temporal differences in DOC concentrations in the text (e.g., between the years and over the vegetation period); it is not clear to me whether the temporal differences are mirrored in the standard deviations shown in Figure 1?"

The temporal variation of the DOC concentration is indeed an interesting topic. Considering that we have five ecosystem fluxes and three different management categories for each of the four years, even a brief summary encompasses a lot of additional data. Adding this information to the result section without further discussion (because we agree, it is outside the focus of this manuscript) would only hinder the understanding of the manuscript. We plan to address this topic in a separate manuscript.

Please also note the supplement to this comment:
https://www.biogeosciences-discuss.net/bg-2018-478/bg-2018-478-AC1-
supplement.pdf

—————————————————————

[Figure]

Fig. 1. Figure 2: Percentage reduction of DOC concentrations between topsoil (TOP) and sub-
soil leachates (SUB) as a function of carbon saturation of pedogenic Fe- and Al-(hydr)oxides.
For the Hainich sites (t

---

## Author Comment (AC2) · 7 Feb 2019

Dear John Van Stan, thank you for your detailed review of our manuscript. In the following, we hope to adequately address your constructive comments and questions.

"Biodegradability measurement – were the samples spiked with nutrients to achieve N, P >Redfield limitations? All other details of the bioincubation tests look good. But, if we want to test the biolability of the DOC, then it is important to release the microbes from as many common limitations as possible (e.g. the authors set an optimal, controlled temperature: page 6, line 2). As a test of how much DOC is utilizable by microbes, this is a test of DOC quality, not an environmental rate at which one would expect the

DOC to be utilized. Thus, ensuring the microbes are released from nutrient limitations, arguably, should be standard to allow comparison of DOC quality across studies, sites, between research groups, and independent of differences in C:N:P across environments. If this was not done, I recommend the authors briefly discuss the implications (biodegradation of DOC could have been constrained)."

No additional nutrients were added to our incubation experiment. It was, however, possible to check concentrations of nitrogen (total inorganic N) and phosphorus (ortho-P) in the samples prior pooling for the incubation test.

Calculating maximum nutrient demands for the consumed carbon in our samples by using values for bacterial growth requirement of N and P suggested by Fellman et al. 2008b (40 $\mu$g N l-1 and 8 $\mu$g P l-1 to satisfy growth requirements using a bacterial growth efficiency of 0.4 and a bacterial molar ratio for C:N of 10 and C:P of 50), we found that for throughfall (TF) and litter leachate (LL) samples, constrained biodegradation due to nutrient limitation is not likely. Low concentrations of phosphorous in stemflow (SF) samples may have limited biological degradation.

We will expand the discussion paragraph at Page 17 Line 20 as followed.:

Beside other factors, nutrient availability can affect biological degradation in samples of ecosystem fluxes. In our study, no additional nutrients were added to compensate for possible limitations. We calculated maximum nutrient demands for the consumed carbon in our samples by using values for bacterial growth requirement of nitrogen and phosphorous suggested by Fellman et al. (2008b) and measured concentrations of N and P in the solution samples prior to pooling for the incubation experiment. The results suggested that constrained biodegradation of DOM due to nutrient limitation in TF and LL samples was not likely. Low concentrations of phosphorous in SF samples may, however, have limited biological degradation and the potential %BDOC could be higher than measured, thus even increasing the difference in the biodegradability of DOM between the samples of SF and those of TF and LL.

We will add the table you can find in the supplement to this reply as new Table S5

"There are no measurements/estimates/tests of soil geochemical interactions with infiltrating DOM. As indicated above, biodegradation will likely be limited in natural settings (compared to the bioincubation tests – even for bioincubation tests without the nutrient spiking). I noticed that the other reviewer also believed this to be a shortcoming of the manuscript. As gathering more data along this vein would be difficult (and is, of course, not possible for storms already past unless it was collected at the time), I recommend the other reviewers' solution: provide more discussion of geochemical controls over DOM processes within soils. Perhaps the discussion could have subsections dedicated to biological factors and geochemical factors."

As this comment is similar to referee 1 we give the same answer.

Being part of the DFG priority program 1374 offered the unique opportunity to access various kinds of information about the study sites. Therefore, it was possible to add additional geochemical information like soil texture and elemental composition. We will add the new table S2 you can find in the supplement to this reply to the supporting information.

To assess whether geo-chemical processes controlled the DOM quality in our study,we additionally examined the relation between changes in DOC concentration and theratio between organic carbon content of the mineral soil and the sum of its oxalate-extractable Fe- and Al-content (OC/[Feo+Alo]) and compared the results with findingsof Kindler et al. (2011).

We will ad Fig.1 of these reply as new Fig.2 to the manuscript

caption: Figure 2: Percentage reduction of DOC concentrations between topsoil (TOP) and subsoil leachates (SUB) as a function of carbon saturation of pedogenic Fe- and Al-(hydr)oxides. For the Hainich sites (this study), the reduction of DOC concentrations decreased significantly with increasing OC/(Feo+Alo) ratio (reduction = 84% −

34%*OC/(Feo+Alo); p = 0.027, r = 0.86). We found no significant correlation for the Schorfheide sites (this study). The relative increase of DOC concentrations at high OC surface loadings was likely caused by a concentration effect because of evapotranspiration, while surface sorption was negligible. The shown site names refer to Kindler et al. (2011).

We therefore will expand the discussion section as follows:

Page 13 Line 13: In the study of Kindler et al. (2011), the retention of DOC in mineral soil, expressed as percentage reduction of downward DOC flux, was closely related to the ratio between organic carbon concentration of the mineral soil and the sum of its oxalate-extractable Fe and Al content (OC/[Feo+Alo]). This suggests that the DOC retention in mineral soils is governed by the sorption to the surfaces of Fe- and Al-(hydr)oxides. Furthermore, organic matter sorption decreased exponentially with increasing organic matter coverage of the hydroxide surfaces (Kindler et al. 2011), which suggests that these hydroxides have a limited DOC sorption capacity (our hypothesis ii). In contrast to the findings of Kindler et al. (2011), we compare DOC concentrations, not fluxes. In order to test whether the data of our study fit the findings of Kindler et al. (2011), we plotted changes in DOC concentrations reported by Kindler et al. (2011) together with the data of this study against the ratio of OC/(Feo+Alo) in one graph (Figure 2). Different from fluxes, which always decreased with increasing soil depth in the Kindler et al. (2011) study, DOC concentrations increased with increasing depth at the Hainich sites with the highest OC/(Feo+Alo) ratios of all study regions (Figure 2). This increase in concentrations can be explained by a concentration effect because of evapotranspiration, if the DOC sorption capacity of pedogenic Fe- and Al-(hydr)oxides is saturated. Overall, the retention of DOC in the Hainich soils of this study fitted well to the DOC retention in the European data set of Kindler et al. (2011), who showed that the regional variation in DOC retention can be as large as the variation at continental scale. The concurrent decrease of DOC concentrations and increase of OC/( Feo+Alo) ratios between TOP and SUB (p = 0.027; Figure 2), corroborates the hypothesis that

sorption to pedogenic Fe- and Al-(hydr)oxides controls DOC retention in mineral soils (Kindler et al. 2011). However, the results for mineral soils of the Schorfheide sites did not meet this pattern, as DOC concentrations decreased from TOP to SUB by 33-72% regardless of the OC/( Feo+Alo) ratio (Figure 2). At the Schorfheide sites, other processes than sorption to Fe- and Al-(hydr)oxide surfaces likely governed DOC retention. The Schorfheide soils developed from fluvioglacial quartzitic sands covering carbonate-free glacial till. Because of their poor pH buffering capacity, these soils were very acidic ($pH_{CaCl2} = 3.0 - 3.6$ in topsoils). The mean pH values in soil water samples of the Schorfheide sites was 4.5 in TOP solutions, increasing to 5.5 in SUB solutions. This means that Al-(hydr)oxides were dissolved in the Schorfheide topsoils, increasing $Al^{3+}$-concentrations in soil water and leachates. The pH increase to 5.5 along the way from TOP to SUB likely induced a re-precipitation of Al. We assume that dissolved organic matter transported from TOP to SUB co-precipitated together with $Al^{3+}$ as described by Nierop et al. (2002) and Jansen et al. (2003, 2005) for acidic sandy soils from the Netherlands. If DOM was immobilized as insoluble metal-organic matter precipitate in B horizons, sorption sites on surfaces of pedogenic (hydr)oxides may be less important.

The following references will be added to the manuscript:

Jansen, B., Nierop, K.G.J., and Verstraten, J.M.: Mobility of Fe(II), Fe(III) and Al in acidic forest soils mediated by dissolved organic matter: influence of solution pH and metal/organic carbon ratios. Geoderma, 113, 323– 340, 2003.

Jansen, B., Nierop, K.G.J., and Verstraten, J.M.: Mechanisms controlling themobility of dissolved organic matter, aluminium and iron in podzol B horizons. Eur. J. Soil Sci., 56, 537–550. doi: 10.1111/j.1365-2389.2004.00686.x, 2005.

Nierop, K.G.J., Jansen, B., and Verstraten, J.M.: Dissolved organic matter, aluminium and iron interactions: precipitation induced by metal/carbon ratio, pH and competition. Sci. Total. Environ., 300, 201–211, 2002.

"The meaning of DOM "origin" is unclear. For example, in the abstract: "strong significant effects of origin of ecosystem fluxes" – what is the "origin"? (A) Is it the first contact between precipitation and terrestrial surfaces (in the tree canopy), thus species specific throughfall v. stemflow v. litter leachate? Or, (B) Is it the origin of specific DOM fluorophores/molecular formulas? If (per A) the "origin" variable is used to indicate the initial DOM-enrichment process - throughfall or stemflow or litter leachate (for gap throughfall) - how is this different from the "species" variable? If the "origin" variable is used to indicate the origination of specific indicator fluorophores (like the component C1, "humic-like with terrestrial origin") or FTICR-MS formulas (like the N-rich organic compounds assumed to have atmospheric origins [p. 15, lines 10-15]), then this should be explicitly defined."

With 'DOM origin' we mean the location where the sample was taken, thus meaning the different ecosystem fluxes in our study (species independent throughfall, stemflow, litter leachate, topsoil and subsoil solution). We will define this more precisely in the manuscript.

The changes will be as follows:

Page2 Line2: The DOM concentration and properties along the water flow path through forest ecosystems depend on the location of sampling and hence the type of sampled ecosystem flux, reflecting production and consumption of DOM during the ecosystem passage on transformation processes.

Page2 Line10: Multivariate statistics revealed strong significant effects of ecosystem fluxes and smaller effects of main tree species on DOM quality.

Page12 Line11: We found a significant effect of ecosystem fluxes on DOM composition variables including SUVA254 and %PARAFAC components (PERMANOVA, p = 0.001) explaining 67 % of sample variance.

The term 'origin' associated with the description of fluorescence components and

FT-ICR MS formulas is adopted from the literature and we will keep it through the manuscript.

"Although there is little literature covering throughfall and stemflow DOM quality, the authors missed some studies. Normally, one cannot cite all the studies on a particular topic; however, in this case, since so few studies exist, I recommend their inclusion. Please note that, for one of these papers, I am the lead author and it is not my intention to push my own work, only to account for the few studies on the topic. Introduction and discussion: Throughfall and stemflow DOM concentration, flux and quality (including potential sources and fates) have been reviewed and evaluated by Van Stan & Stubbins, 2018, https://doi.org/10.1002/lol2.10059. Page 16, lines 26-30: The authors only reference Qualls & Haines (1992) biodegradation estimates for throughfall. But, they do not discuss the only study reported stemflow BDOC in Qualls' recent Special Issue (Howard et al., 2018, https://doi.org/10.3390/f9050236)."

Thank you, for pointing us at these studies, which we indeed missed. We will include information from van Stan and Stubbins (2018) as reference in the introduction section of our manuscript (Page 3 Line 2). We will include the results of Howard et al. (2018) in the discussion section (Page 16 Line 30) as follows:

With a cumulative degradation of up to 40 % of the initial DOC concentration as well as the highest degradations rates, DOM from SF samples was most bioavailable. TF samples with BDOC up to 36 % contained DOM, which seemed slightly less bioavailable. This is within the results from Howard et al. (2018) reporting BDOC in an interquartile range of 36-73% for cedar throughfall and stemflow samples.

Both references will be added to the reference list.

Please also note the supplement to this comment: https://www.biogeosciences-discuss.net/bg-2018-478/bg-2018-478-AC2-supplement.pdf
* * *
[Figure]

**Fig. 1.** Figure 2: Percentage reduction of DOC concentrations between topsoil (TOP) and subsoil leachates (SUB) as a function of carbon saturation of pedogenic Fe- and Al-(hydr)oxides. For the Hainich sites (t

---

## Author Comment (AC3) · 7 Feb 2019

Dear Malak Tfaily, thank you for your detailed review of our manuscript. In the following, we hope to adequately address your constructive comments and questions.

"Page 5, line 31, did you check if 0.2 $\mu$m was enough to eliminate microbial communities originally present in the samples?"

We did not check for complete bacteria removal but filtration through 0.2 $\mu$m is a standard procedure for the removal of microorganisms. To our knowledge already a filter pore diameter of < 1 $\mu$m would be sufficient to exclude living bacterial cells so that

only even smaller spores could have passed. Please note that the samples were subsequently (re-)inoculated with a microbial community extracted from the soils of the biodiversity exploratory. Therefore, the goal of the filtration was not necessarily a complete sterilization, but a standardization of the microbial community degrading the DOM in the different samples.

"Pages 4-5 what was the total number of samples and how was it distributed in terms of management? Line 16, can you give the break down for the 466 samples?"

We think, the number of analyzed samples per measurement/experiment you are asking for is given in Table 1. Detailed information of pooled samples for FT-ICR-MS and the incubation test regarding management distribution are given in Tables S3 and S4 in the supplement. Moreover, in the supplement (Table S2), detailed numbers of samples per site, plot and ecosystem flux for the 466 fluorescence samples are given.

"Line 16, what do you mean by: To balance uneven sample numbers, we calculated mean EEMs per plot and ecosystem flux resulting in a dataset with 79 samples. Did you collapse the 466 samples into 79 samples to allow for plot versus plot comparison? what was the variability within the same plot? Line 20, how did the optical data look for these samples? "

As is visible in Table S2 the number of available sample numbers per plot and ecosystem flux was not the same for the various sample types (n>10 for TF, SF and LL; n<5 for TOP and SUB). In their tutorial review to PARAFAC, Murphy et al. (2013) caution about unequal numbers of replicated samples in PARAFAC modeling. To avoid this influence, we calculated mean EEMs to use one 'sample' per plot and ecosystem flux to gain a representative model. To give you an impression, Fig.1 shows throughfall (TF) EEM plots for the Schorfheide plot SEW8 (beech unmanaged) for different sampling dates and the resulting mean EEM used in our statistical analysis.

"Do you believe that differences due to management is higher than that between plots within the same forest?"

Detecting differences in fluorescence spectra caused by different management practice versus intra-plot variability depends on the management categories compared. With fluorescence measurements of DOM (please keep in mind that fluorescence measurements address only the portion of DOM able to absorb and emit light) we were able to detect differences due to management decisions like tree species selection. Possible differences between differently managed forests with the same tree species, in our case unmanaged and age-class beech forests, were not statistically distinguishable.

"Page 7, line 14, only six spectra were averaged? Typically, we do at least 100."

Our samples were measured on a FT-ICR-MS Ultra (ThermoFisher equipped with the SIMION optimized ICR cell for more homogeneous magnetic field in contrast to the first ThermoFisher edition, resulting in the specified exactness better than 1ppm). Our own working group-intern improvements led to an exactness better than 500 ppb (as example, mean deviation in this data set is 400 ppb), today. For previous studies in another institute, we used an Apex II Bruker (but of course not for DOM measurements), therefore we know some important technical differences between the ThermoFisher and the Bruker MS, which are important for the comparison of spectra from different instrument types. The need to average large numbers of at least 100 scans is probably related to (I) the use of different mass spectrometers, namely of the suppliers Bruker and Thermo Finnigan, and (II) same term (scan) for different things. The Bruker machine allows to accumulate scans without limit, which means, the longer you measure, the more intensive your mass peaks become (therefore, most people sum up 100-200 scans). In contrast, with Thermo instruments only an accumulation up to 50 so-called $\mu$-scans is noticeable, more $\mu$-scans do not change the signal intensity (kind of included system averaging without any possibility for changing or even excluding by the operator). Each of these 50 $\mu$-scans is one transient, 50 $\mu$-scans together are $\sim$ 3 min measuring time and were combined subsequently to one so-called scan. For further improving spectra quality, we average (no accumulation possible) the data of six such Thermo-scans (with 50 $\mu$-scans each, in total recording time $\sim$ 18 min).

[Figure]

As an example result for a beech throughfall sample we got 18010 peaks in the averaged spectrum of 6 scans, but it would be only 13003 peaks with 4 scans - all with the same intensive peaks from first scan on. (100 of such scans would imply 300 min = 5 h).

"Can you provide more details regarding formula assignment and the rules that were used? What was the number of unassigned formula? What were the ranges of C, H, N, C, O, S, P? that were used?"

Thank you for pointing out the missing ranges of formula assignments, who will be added to the manuscript.

After manual examinations of included heteroatoms the following constraints were chosen to generate empirical formulae from all peaks: C, H and O unlimited, N and S: 0-3 (without the combination S>1N3), 13C: 0-1 and P=0.

Rules (from organic mass spectrometry): for odd numbered peaks: N=0 or 2 (nitrogen rule), no 13C

For even numbered peaks: no 12CHO compounds; N=1 or 3 or 13C=1, but not in combination

At least O2 incorporated (that means, one COOH group for neg. ions), that means no pure CH compounds

$H/C \leq 2.0$

$O/C \leq 1.4$

Because of only small peaks for CHOSN, we decided to show van Krevelen plots for CHO compounds only, in contrast to other projects. An example of formula numbers of beech throughfall sample can be found in the supporting information to this reply.

"Figure 2, it's hard to see the zoom in but be careful about peak splitting as this can affect your formula assignment. Even though it is hard to see, it appears you had some

peak splitting."

Independent of daily tuning and calibrating, in the course of instrument life time, the peak shapes are changing (quenching of peaks after detections lead to slow adsorption of chemicals on the ICR cell walls too). For best conditions in very complex mixtures like DOM, we have got a (company developed) procedure for adjustment of ICR cell parameters (as result replacing the instrument master file). This procedure is usually not distributed to customers, because of the possibility to misalignments. The procedure is measuring defined standards the first two runs automatically ($\sim$ 1 day), followed by a manual adjustment, especially under consideration of the peak legs (should achieved < 15% peak height ). That means, when the peak legs increase (independent of fresh cleaning of the ion source & quadrupole 0), we adjust the cell parameters to prevent peak splitting. You can see randomly selected nominal masses from two different samples as an example in Fig. 2 and Fig.3. In both examples, assignments by our calculation program and manually are highlighted. Also impossible assignments due to multiple possibilities and/or limitation of our constraints or just because of decreasing exactness of peaks due to very small intensities are indicated. As you may see, for all peaks there is at least an explanation – and we found no signs for peak splitting or peaks without sum formula proposal. As you may see, at our Thermo instrument the base line is not visible – obviously base corrected by the manufacturer. We assume, this is the reason for missing peaks bases, which could led to misinterpretations.

Non the less, the figures included in the discussion manuscript were converted to pdf with the whole document. For the final manuscript all figures will be submitted separately as individual files hopefully avoiding resolution losses.

Please also note the supplement to this comment:
https://www.biogeosciences-discuss.net/bg-2018-478/bg-2018-478-AC3-supplement.pdf

**SEW8–TF  beech unmanaged**

**Fig. 1.** Fluorescence EEMs plots of different throughfall samples of Schorfheide plot SEW8 (beech unmanaged) for different sampling dates. Mean= mean EEM calculated of all shown measurements

**Beech, throughfall (m/z 485)**

Thieme_14_Recal #2-7  RT: 2.74-16.17  AV: 6  NL: 4.47E2
T: FTMS - p ESI Full ms [200.00-1000.00]

**Scilab based assignment**

| nr. | mass | intensity | m-diff. | C | H | O | N | S | P | DBE | NOSC | AI |
|---|---|---|---|---|---|---|---|---|---|---|---|---|
| 1 | 485.019700 | 38.500 | 0.426 | 23. | 17. | 6. | 0. | 3. | 0. | 16.0 | 0.0 | 0.4 |
| 2 | 485.030260 | 49.400 | -0.030 | 29. | 9. | 8. | 0. | 0. | 0. | 26.0 | 0.2 | 0.8 |
| 3 | 485.036150 | 448.100 | -0.013 | 22. | 13. | 13. | 0. | 0. | 0. | 17.0 | 0.6 | 0.3 |
| 4 | 485.055200 | 43.000 | 0.410 | 23. | 17. | 10. | 0. | 1. | 0. | 16.0 | 0.2 | 0.3 |
| 5 | 485.057500 | 245.300 | 0.208 | 19. | 17. | 15. | 0. | 0. | 0. | 12.0 | 0.7 | -1.0 |
| 6 | 485.061260 | 123.500 | -0.272 | 17. | 25. | 10. | 0. | 3. | 0. | 6.0 | 0.1 | -2.0 |
| 7 | 485.067100 | 103.700 | 0.424 | 30. | 13. | 7. | 0. | 0. | 0. | 25.0 | 0.0 | 0.7 |
| 8 | 485.070200 | 37.900 | 0.153 | 27. | 17. | 7. | 0. | 1. | 0. | 20.0 | -0.0 | 0.6 |
| 9 | 485.072780 | 118.200 | 0.231 | 23. | 17. | 12. | 0. | 0. | 0. | 16.0 | 0.3 | 0.3 |
| 10 | 485.091760 | 150.200 | -0.285 | 25. | 25. | 4. | 0. | 3. | 0. | 14.0 | -0.4 | 0.3 |
| 11 | 485.093680 | 372.500 | 0.002 | 20. | 21. | 14. | 0. | 0. | 0. | 11.0 | 0.4 | -0.7 |
| 12 | 485.123500 | 55.700 | -0.038 | 20. | 25. | 10. | 2. | 1. | 0. | 10.0 | 0.1 | -0.4 |
| 13 | 485.128120 | 17.900 | -0.310 | 26. | 29. | 3. | 0. | 3. | 0. | 13.0 | -0.7 | 0.3 |
| 14 | 485.130180 | 54.800 | 0.117 | 21. | 25. | 13. | 0. | 0. | 0. | 10.0 | 0.0 | -0.5 |
| 15 | 485.133630 | 74.600 | 0.196 | 18. | 29. | 13. | 0. | 1. | 0. | 5.0 | -0.1 | -2.5 |
| 16 | 485.157890 | 67.700 | 0.468 | 25. | 29. | 4. | 2. | 2. | 0. | 13.0 | -0.4 | 0.3 |
| 17 | 485.166480 | 105.200 | 0.031 | 22. | 29. | 12. | 0. | 0. | 0. | 9.0 | -0.2 | -0.4 |

CHO compounds in green
CHOS$_x$ compounds in red
CHON$_y$ compounds in blue
CHON$_y$S$_x$ compounds in brown

* not assigned by limitation of the constraints (e.g. N>3)
  or multiple assignments

**Fig. 2.** Randomly selected nominal mass from beech throughfall sample

**pine, litter leachate (m/z 485)**

CHO compounds in green
CHOS$_x$ compounds in red
CHON$_y$ compounds in blue
CHON$_y$S$_x$ compounds in brown

* not assigned by limitation of the constraints
(e.g. N>3) or multiple assignments

| 1 | 485.000460 | 55.100 | -0.187 | 22. | 13. | 9. | 0. | 2. | 0. | 17.0 | 0.4 | 0.5 |
| 2 | 485.025900 | 54.100 | -0.367 | 24. | 9. | 10. | 2. | 0. | 0. | 22.0 | 0.7 | 0.8 |
| 3 | 485.034190 | 90.600 | -0.340 | 27. | 17. | 3. | 0. | 3. | 0. | 20.0 | -0.2 | 0.6 |
| 4 | 485.036360 | 312.000 | 0.197 | 22. | 13. | 13. | 0. | 0. | 0. | 17.0 | 0.6 | 0.3 |
| 5 | 485.040550 | 49.200 | 0.147 | 20. | 21. | 8. | 0. | 3. | 0. | 11.0 | 0.0 | -0.1 |
| 6 | 485.055610 | 231.800 | -0.049 | 24. | 21. | 5. | 0. | 3. | 0. | 15.0 | -0.2 | 0.4 |
| 7 | 485.057290 | 956.800 | -0.002 | 19. | 17. | 15. | 0. | 0. | 0. | 12.0 | 0.7 | -1.0 |
| 8 | 485.072380 | 198.200 | -0.169 | 23. | 17. | 12. | 0. | 0. | 0. | 16.0 | 0.3 | 0.3 |
| 9 | 485.077660 | 80.300 | -0.249 | 29. | 13. | 6. | 2. | 0. | 0. | 25.0 | 0.2 | 0.8 |
| 10 | 485.088510 | 89.500 | -0.164 | 28. | 21. | 4. | 0. | 2. | 0. | 19.0 | -0.3 | 0.5 |
| 11 | 485.091330 | 169.000 | 0.154 | 24. | 21. | 9. | 0. | 1. | 0. | 15.0 | -0.0 | 0.3 |
| 12 | 485.093670 | 1255.200 | -0.008 | 20. | 21. | 14. | 0. | 0. | 0. | 11.0 | 0.4 | -0.7 |
| 13 | 485.097370 | 46.100 | 0.321 | 17. | 25. | 14. | 0. | 1. | 0. | 6.0 | 0.3 | -5.0 |
| 14 | 485.109060 | 327.100 | 0.126 | 24. | 21. | 11. | 0. | 0. | 0. | 15.0 | 0.0 | 0.2 |
| 15 | 485.112140 | 170.100 | -0.165 | 21. | 25. | 11. | 0. | 1. | 0. | 10.0 | -0.0 | -0.3 |
| 16 | 485.124470 | 42.600 | 0.280 | 28. | 21. | 8. | 0. | 0. | 0. | 19.0 | -0.2 | 0.5 |
| 17 | 485.130020 | 1051.700 | -0.043 | 21. | 25. | 13. | 0. | 0. | 0. | 10.0 | 0.0 | -0.5 |
| 18 | 485.138990 | 49.200 | -0.456 | 32. | 21. | 5. | 0. | 0. | 0. | 23.0 | -0.3 | 0.6 |
| 19 | 485.145020 | 89.900 | -0.299 | 25. | 25. | 10. | 0. | 0. | 0. | 14.0 | -0.2 | 0.2 |
| 20 | 485.150000 | 40.000 | 0.441 | 23. | 33. | 5. | 0. | 3. | 0. | 8.0 | -0.7 | -0.1 |
| 21 | 485.166410 | 950.300 | -0.039 | 22. | 29. | 12. | 0. | 0. | 0. | 9.0 | -0.2 | -0.4 |
| 22 | 485.181680 | 140.500 | -0.025 | 26. | 29. | 9. | 0. | 0. | 0. | 13.0 | -0.4 | 0.2 |
| 23 | 485.187400 | 52.200 | -0.178 | 19. | 33. | 14. | 0. | 0. | 0. | 4.0 | -0.3 | -2.2 |
| 24 | 485.193690 | 66.900 | -0.117 | 26. | 33. | 3. | 2. | 2. | 0. | 12.0 | -0.7 | 0.3 |
| 25 | 485.200160 | 104.400 | -0.172 | 27. | 33. | 6. | 0. | 1. | 0. | 12.0 | -0.7 | 0.2 |
| 26 | 485.202930 | 435.500 | 0.096 | 23. | 33. | 11. | 0. | 0. | 0. | 8.0 | -0.5 | -0.3 |

**Fig. 3.** Randomly selected nominal mass from pine litter leachate sample

**Supplement:**

Example of number_of_formulas for beech throughfall

| | Number of formulas | comments |
|---|---|---|
| Registered peaks: | 18010 | |
| Unique identified peaks | 9878 | using the constraints above mentioned in the reply |
| Multiple identification | 2160 | These were not further considered, that means, we lose 12% of the peaks, but we exclude false positive ones |
| Not listed peaks | 5972 | most of all are $^{13}$C isotopic peaks without further scientific information; or not assigned because not within constraints, e.g. $CHOP$, $CHOS_4$, $CHOS_2N_3$, $CHOS_3N_3$ …; or the peaks are so small, that their exactness is smaller than 1 ppm |

---

## Author Comment (AC4) · 7 Feb 2019

We noticed the caption of Fig.1 in our reply was incomplete. Please find here the complete caption.

Fig.1: Percentage reduction of DOC concentrations between topsoil (TOP) and subsoil leachates (SUB) as a function of carbon saturation of pedogenic Fe- and Al-(hydr)oxides. For the Hainich sites (this study), the reduction of DOC concentrations decreased significantly with increasing OC/(Feo+Alo) ratio (reduction = 84% − 34%*OC/(Feo+Alo); p = 0.027, r = 0.86). We found no significant correlation for the Schorfheide sites (this study). The relative increase of DOC concentrations at high OC

[Figure]

surface loadings was likely caused by a concentration effect because of evapotranspiration, while surface sorption was negligible. The shown site names refer to Kindler et al. (2011).

---

## Author Response (AR1)

**Reply to Referee 1**

Thank you for your detailed review of our manuscript. In the following, we hope to adequately address your constructive comments and questions.

*Yet, one major shortcoming of the manuscript is that the properties of the soils of the study sites are not well described. There is some information about parent material and soil type given in the supplementary material, however, in my opinion a study on DOM in subsoil requires more detailed information about soil properties that may strongly affect the movement of DOM such as soil texture, mineral composition of the reactive clay fraction and pH (i.e., properties known to determine the chemistry of sorption processes). Hence, the focus of the discussion is a bit too much on biological factors of DOM movement, while the geo-chemical controls of soil processes should be covered a bit more. In case the data on soil mineralogy and chemistry are not available, I suggest to (at least) enhance the discussion on basis of available literature about how differences or similarities in geo-chemical factors between sites might have influenced the results of the present study. For instance, a decrease in highly oxidized compounds (page 15, line 21) might be explained by binding of the carboxyl-groups to positively charged surfaces of Al-/Fe-oxides at acidic pH values. In my view, the conclusion that the alignment of the composition of DOM is due to biotransformation and "interaction with the soil solid phase" needs support by considerations about such processes*

Being part of the DFG priority program 1374 offered the unique opportunity to access various kinds of information about the study sites. Therefore, it was possible to add additional geochemical information like soil texture and elemental composition.

We will add the following table to the manuscript as new Table 1:

**Table 1: Chemical soil properties and mean dissolved organic carbon (DOC) concentrations of plots in the Hainich Dün (HEW) and Schorfheide Chorin (SEW) sites. LL = litter leachate, TOP = topsoil, SUB = subsoil, reduction cDOC (%) = reduction of DOC concentration in % between LL and TOP or TOP and SUB, Corg = organic carbon content of soil, $Al_0$ = aluminum content extracted with ammonium oxalate, $Fe_0$ = iron content extracted with ammonium oxalate**

| plot | ecosystem flux / soil layer | management category | DOC [mg/L] | reduction cDOC [%] | $C_{org}$ [g/kg] | $Al_o$ [g/kg] | $Fe_o$ [g/kg] | clay [g/kg] | texture (KA5*) | pH soil (CaCl2) |
|---|---|---|---|---|---|---|---|---|---|---|
| HEW1 | LL | category | 39.23 | | | | | | | |
| HEW1 | Top | coniferous age-class | 11.26 | 71.31 | 69.14 | 3.28 | 3.50 | 326 | Lu | 7.0 |
| HEW1 | Sub | coniferous age-class | 15.46 | -37.36 | 28.99 | 4.38 | 3.89 | 239 | Uls/Tl | 7.5 |
| HEW2 | LL | coniferous age-class | 41.54 | | | | | | | |
| HEW2 | Top | coniferous age-class | 24.72 | 40.49 | 50.60 | 1.43 | 4.92 | 241 | Lu /Ut4 | 4.6 |
| HEW2 | Sub | coniferous age-class | 7.70 | 68.84 | 6.95 | 1.73 | 2.98 | 589 | Tu2 | 7.0 |
| HEW3 | LL | coniferous age-class | 66.08 | | | | | | | |
| HEW3 | Top | coniferous age-class | 16.76 | 74.64 | 47.74 | 2.33 | 3.18 | 359 | Ut3/Ut2 | 3.9 |
| HEW3 | Sub | coniferous age-class | 14.04 | 16.22 | 10.33 | 2.38 | 2.37 | 634 | Tl | 6.7 |
| HEW4 | LL | deciduous age-class | 22.76 | | | | | | | |
| HEW5 | LL | deciduous age-class | 18.26 | | | | | | | |
| HEW5 | Top | deciduous age-class | 7.50 | 58.92 | 61.77 | 3.79 | 3.19 | 457 | Lu | 5.2 |
| HEW5 | Sub | deciduous age-class | 5.12 | 31.78 | | | | | | 7.2 |
| HEW6 | LL | deciduous age-class | 17.57 | | | | | | | |
| HEW6 | Top | deciduous age-class | 11.30 | 35.70 | 34.40 | 2.19 | 3.73 | 214 | Lu | 4.3 |
| HEW6 | Sub | deciduous age-class | 5.20 | 54.02 | 5.15 | 2.45 | 3.62 | 442 | Tu2/Tl | 5.4 |
| HEW10 | LL | unmanaged | 24.18 | | | | | | | |
| HEW10 | Top | unmanaged | 7.95 | 67.15 | 67.59 | 3.49 | 4.74 | 485 | Ut4 | 4.1 |
| HEW11 | LL | unmanaged | 29.77 | | | | | | | |
| HEW11 | Top | unmanaged | 10.96 | 63.20 | 58.52 | 3.31 | 4.72 | 404 | Ut4 | 4.9 |
| HEW11 | Sub | unmanaged | 12.10 | -10.41 | 19.78 | 3.46 | 4.32 | 517 | Tu3 | 4.9 |
| HEW12 | LL | unmanaged | 24.02 | | | | | | | |
| HEW12 | Top | unmanaged | 7.42 | 69.09 | 31.13 | 1.72 | 2.64 | 164 | Ut4 | 3.9 |
| HEW12 | Sub | unmanaged | 5.60 | 24.52 | 5.58 | 2.43 | 3.19 | 424 | Tu2 | 5.9 |
| SEW1 | LL | coniferous age-class | 67.07 | | | | | | | |
| SEW1 | Top | coniferous age-class | 58.63 | 12.59 | 18.34 | 1.82 | 2.02 | 5 | Sl2 | 3.6 |
| SEW1 | Sub | coniferous age-class | 16.19 | 72.39 | 2.06 | 2.05 | 2.03 | 1 | Sl2 | 3.9 |
| SEW2 | LL | coniferous age-class | 58.50 | | | | | | | |
| SEW2 | TOP | coniferous age-class | 26.73 | 54.31 | 16.99 | 1.78 | 1.94 | 32 | Sl2 | 3.5 |
| SEW2 | Sub | coniferous age-class | 11.40 | 57.34 | 2.26 | 2.68 | 2.50 | 33 | Sl2 | 4.2 |
| SEW3 | LL | coniferous age-class | 57.20 | | | | | | | |
| SEW3 | Top | coniferous age-class | 37.09 | 35.15 | 20.95 | 1.61 | 1.62 | 17 | Sl2 | 3.3 |
| SEW3 | Sub | coniferous age-class | 15.06 | 59.39 | 4.05 | 2.09 | 1.38 | 3 | Sl2 | 4.0 |
| SEW5 | LL | deciduous age-class | 32.84 | | | | | | | |
| SEW5 | Top | deciduous age-class | 91.81 | -179.59 | 29.56 | 1.20 | 1.04 | 1 | Sl2 | 3.1 |
| SEW5 | Sub | deciduous age-class | 27.86 | 69.65 | 2.50 | 2.21 | 1.29 | 1 | Sl2/Su2 | 3.4 |
| SEW6 | LL | deciduous age-class | 37.84 | | | | | | | |
| SEW6 | Top | deciduous age-class | 12.84 | 66.05 | 31.05 | 2.39 | 2.52 | 23 | Sl2 | 3.4 |
| SEW6 | Sub | deciduous age-class | 8.48 | 34.00 | 1.45 | 1.77 | 1.60 | 17 | Sl2 | 3.9 |
| SEW7 | LL | unmanaged | 26.20 | | | | | | | |
| SEW7 | Top | unmanaged | 46.86 | -78.84 | 24.30 | 1.74 | 1.78 | 1 | Sl2 | 3.2 |
| SEW7 | Sub | unmanaged | 16.89 | 63.96 | 6.37 | 1.38 | 1.55 | | Sl2 | 3.7 |
| SEW8 | LL | unmanaged | 41.33 | | | | | | | |
| SEW8 | Top | unmanaged | 29.03 | 29.76 | 29.20 | 1.86 | 1.58 | 20 | Sl2 | 3.1 |
| SEW8 | Sub | unmanaged | 13.07 | 54.97 | 10.28 | 1.52 | 1.48 | 1 | Sl2 | 3.2 |
| SEW9 | LL | unmanaged | 42.50 | | | | | | | |
| SEW9 | Top | unmanaged | 39.94 | 6.01 | 22.96 | 0.95 | 1.01 | 18 | Sl2 | 3.0 |
| SEW9 | Sub | unmanaged | 14.92 | 62.65 | 4.81 | 1.43 | 1.09 | 1 | Sl2 | 3.7 |

* KA5 = Ad-Hoc-Arbeitsgruppe Boden (2005)

To assess whether geo-chemical processes controlled the DOM quality in our study, we additionally examined the relation between changes in DOC concentration and the ratio between organic carbon content of the mineral soil and the sum of its oxalate-extractable Fe- and Al-content ($OC/[Fe_o+Al_o]$) and compared the results with findings of Kindler et al. (2011).

We therefore will expand the discussion section as follows:

Page 13, line 25: In the study of Kindler et al. (2011), the retention of DOC in mineral soil, expressed as percentage reduction of downward DOC flux, was closely related to the ratio between organic carbon content of the mineral soil and the sum of its oxalate-extractable Fe and Al content ($OC/[Fe_o+Al_o]$). This suggested that the DOC retention in mineral soils is governed by the sorption to the surfaces of Fe- and Al-(hydr)oxides. Because Fe- and Al-(hydr)oxides have a limited sorption capacity for organic matter, DOC retention in subsoils decreased exponentially with increasing organic matter coverage of the hydroxide surfacrs (Kindler et al. 2011). In contrast

to the findings of Kindler et al. (2011), we compare DOC concentrations, not fluxes. In order to test whether the data of our study fit the findings of Kindler et al. (2011), we plotted changes in DOC concentrations reported by Kindler et al. (2011) together with the data of this study against the ratio of $OC/(Fe_o+Al_o)$ in one graph (Figure 2). Different from fluxes, which always decreased with increasing soil depth in the Kindler et al. (2011) study, DOC concentrations increased with increasing depth at the Hainich sites with the highest $OC/(Fe_o+Al_o)$ ratios of all study regions (Figure 2). This increase in concentrations can be explained by a concentration effect because of evapotranspiration, in the case that the DOC sorption capacity of pedogenic Fe- and Al-(hydr)oxides is saturated. Overall, the retention of DOC in the Hainich soils of this study fitted well to the DOC retention in the European data set of Kindler et al. (2011), which showed that the regional variation of DOC retention can be as large as the variation at continental scale. The reduction of DOC concentrations between TOP and SUB significantly decreased with increasing $OC/(Fe_o+Al_o)$ ratio ($p = 0.027$; Figure 2), corroborating the hypothesis that sorption to pedogenic Fe- and Al-(hydr)oxides controled DOC retention in mineral soils (Kindler et al. 2011). However, the results for mineral soils of the Schorfheide sites did not follow this pattern, as DOC concentrations decreased from TOP to SUB by 33–72% regardless of the $OC/(Fe_o+Al_o)$ ratio (Figure 2). At the Schorfheide sites, other processes than sorption to Fe- and Al-(hydr)oxide surfaces likely governed DOC retention. The Schorfheide soils developed from fluvioglacial quartzitic sands covering carbonate-free glacial till. Because of their poor pH buffering capacity, these soils were very acidic ($pH_{CaCl2} = 3.0–3.6$ in topsoils). The mean pH value in soil water samples of the Schorfheide sites was 4.5 in TOP solutions, increasing to 5.5 in SUB solutions. This means that Al-(hydr)oxides were dissolved in the Schorfheide topsoils, increasing $Al^{3+}$-concentrations in soil water and leachates. The pH increase to 5.5 along the way from TOP to SUB likely induced a re-precipitation of Al. We assume that dissolved organic matter transported from TOP to SUB co-precipitated together with $Al^{3+}$ as described by Nierop et al. (2002) and Jansen et al. (2003, 2005) for acidic sandy soils from the Netherlands. If DOM was immobilized as insoluble metal-organic matter precipitate in B-horizons, no limitation by available sorption sites of surfaces of pedogenic (hydr)oxides would apply, so that reductions of DOC concentrations with increasing depth in mineral soil would be independent of the soils $OC/(Fe_o+Al_o)$ saturation index.

The following references will be added to the manuscript:

Jansen, B., Nierop, K.G.J., and Verstraten, J.M.: Mobility of Fe(II), Fe(III) and Al in acidic forest soils mediated by dissolved organic matter: influence of solution pH and metal/organic carbon ratios. Geoderma, 113, 323– 340, 2003.

Jansen, B., Nierop, K.G.J., and Verstraten, J.M.: Mechanisms controlling themobility of dissolved organic matter, aluminium and iron in podzol B horizons. Eur. J. Soil Sci., 56, 537–550. doi: 10.1111/j.1365-2389.2004.00686.x, 2005.

Nierop, K.G.J., Jansen, B., and Verstraten, J.M.: Dissolved organic matter, aluminium and iron interactions: precipitation induced by metal/carbon ratio, pH and competition. Sci. Total. Environ., 300, 201–211, 2002.

The following figure will be added to the manuscript as new Figure 2:

[Figure]

**Figure 2: Percentage reduction of DOC concentrations between topsoil (TOP) and subsoil leachates (SUB) as a function of carbon saturation of pedogenic Fe- and Al-(hydr)oxides. For the Hainich sites (this study), the reduction of DOC concentrations decreased significantly with increasing $OC/(Fe_o+Al_o)$ ratio (reduction = 84% – 34%*$OC/(Fe_o+Al_o)$; p = 0.027, r = 0.86). We found no significant correlation for the Schorfheide sites (this study). The relative increase of DOC concentrations at high OC surface loadings was likely caused by a concentration effect because of evapotranspiration, while surface sorption was negligible. The shown site names refer to Kindler et al. (2011).**

*Page 3, Line 27: The hypotheses could be stated more precisely, i.e., currently a broad predicition is made ("DOM changes systematically") without any consideration about the main mechanisms. How and why should the composition and biodegradability of DOM change along the water pathway? How and why should tree species and forest management affect these changes?*

To state our hypotheses more precisely we will change the manuscript as follows:

Page 3, line 27: We hypothesized *i*) that the composition and the biodegradability of DOM changes from a dominance of non-aromatic nitrogen-rich compounds of high bioavailability to highly aromatic, increasingly-oxidized, nitrogen-poor compounds with decreased bioavailability along the water flow path through forest ecosystems, from throughfall (TF), stemflow (SF), litter layer leachate (LL) to mineral topsoil (TOP) and subsoil (SUB) solution. We postulated *ii*) that aboveground changes of DOC concentrations and DOM composition are mainly controlled by selective biological degradation, while changes in mineral soil are governed by sorption to mineral surfaces. Moreover, we hypothesized *iii*) that the dominant tree species as well as forest management intensity affect the DOM composition as well as the direction and magnitude of its changes along the water flow path. The former because of the presence of species-specific compounds in DOM, like phenolic secondary metabolites in beech forests, the latter, measured as the Forest Management Index (ForMI), beside others because of its influence on the biomass production and C input into the soil (Kahl and Bauhus, 2014).

*Page 3, Line 30: I suggest to briefly explain the ForMI here so that the readers can gain a better understanding of the study approach.*

The ForMI is explained now in more detail on page 4, lines 18ff of the revised manuscript.

*Page 4, Line 17: The work of Fischer et al. is not given in the reference list.*

Thank you for pointing this out. The reference will be added in the reference section

*Results section: Although it is not the focus of the manuscript, it may be interesting to briefly summarize the magnitude of the temporal differences in DOC concentrations in the text (e.g., between the years and over the vegetation period); it is not clear to me whether the temporal differences are mirrored in the standard deviations shown in Figure 1?*

The temporal variation of the DOC concentration is indeed an interesting topic. Considering that we have five ecosystem fluxes and three different management categories for each of the four years, even a brief summary encompasses a lot of additional data. Adding this information to the result section without further discussion (because we agree, it is outside the focus of this manuscript) would only hinder the understanding of the manuscript. We plan to address this topic in a separate manuscript.

**Reply to Referee 2**

Dear John Van Stan, thank you for your detailed review of our manuscript. In the following, we hope to adequately address your constructive comments and questions.

*Biodegradability measurement – were the samples spiked with nutrients to achieve N, P >Redfield limitations? All other details of the bioincubation tests look good. But, if we want to test the biolability of the DOC, then it is important to release the microbes from as many common limitations as possible (e.g. the authors set an optimal, controlled temperature: page 6, line 2). As a test of how much DOC is utilizable by microbes, this is a test of DOC quality, not an environmental rate at which one would expect the DOC to be utilized. Thus, ensuring the microbes are released from nutrient limitations, arguably, should be standard to allow comparison of DOC quality across studies, sites, between research groups, and independent of differences in C: N: P across environments. If this was not done, I recommend the authors briefly discuss the implications (biodegradation of DOC could have been constrained).*

No additional nutrients were added to our incubation experiment. It was, however, possible to check concentrations of nitrogen (total inorganic N) and phosphorus (ortho-P) in the samples prior pooling for the incubation test. We will add the following table to the supplement as new Table S5

**Table S5: sample information: DOM biodegradability. Mean concentrations of nitrogen and phosphor in samples before incubation. $NH_4$-N + $NO_3$-N = concentration of ammonium and nitrate nitrogen, $PO_4$-P = concentration of ortho phophate**

| | TF | | SF | | LL | |
|---|---|---|---|---|---|---|
| | $NH_4$-N + $NO_3$-N [mg L$^{-1}$] | $PO_4$-P [mg L$^{-1}$] | $NH_4$-N + $NO_3$-N [mg L$^{-1}$] | $PO_4$-P [mg L$^{-1}$] | $NH_4$-N + $NO_3$-N [mg L$^{-1}$] | $PO_4$-P [mg L$^{-1}$] |
| *Schwäbische Alb* | | | | | | |
| coniferous age-class | 1.42 | 0.047 | | | 3.85 | 0.206 |
| deciduous age-class | 1.02 | 0.144 | 4.37 | 0.830 | 0.82 | 0.184 |
| unmanaged | 1.45 | 0.052 | 3.44 | 0.136 | 5.14 | 0.349 |
| *Hainich Dün* | | | | | | |
| coniferous age-class | 7.39 | 0.330 | 18.68 | 0.086 | 4.95 | 0.113 |
| deciduous age-class | 2.97 | 0.146 | 2.24 | 0.008 | 5.23 | 0.828 |
| unmanaged | 2.44 | 0.124 | 2.19 | 0.017 | 7.17 | 1.158 |
| *Schorfheide Chorin* | | | | | | |
| coniferous age-class | 0.92 | 0.018 | 6.57 | 0.015 | 0.77 | 0.301 |
| deciduous age-class | 0.71 | 0.232 | 1.34 | 0.124 | 7.04 | 0.323 |
| unmanaged | 2.27 | 0.727 | 1.25 | 0.130 | 1.97 | 0.504 |

Calculating maximum nutrient demands for the consumed carbon in our samples by using values for bacterial growth requirement of N and P suggested by Fellman et al. 2008b (40 µg N l$^{-1}$ and 8 µg P l$^{-1}$ to satisfy growth requirements using a bacterial growth efficiency of 0.4 and a bacterial molar ratio for C:N of 10 and C:P of 50), we found that for throughfall (TF) and litter leachate (LL) samples, constrained biodegradation due to nutrient limitation is not likely. Low concentrations of phosphorous in stemflow (SF) samples may have limitedbiological degradation.

We will expand the discussion paragraph at Page 17, line 20 as followed.:

Beside other factors, nutrient availability can affect biological degradation in samples of ecosystem fluxes. In our study, no additional nutrients were added to compensate for possible limitations. We calculated maximum nutrient demands for the consumed carbon in our samples by using values for bacterial growth requirement of nitrogen and phosphorous suggested by Fellman et al. (2008b) and measured concentrations of N and P in the solution samples prior to pooling for the incubation experiment. The results suggested that constrained biodegradation of DOM due to nutrient limitation in TF and LL samples was not likely. Low concentrations of phosphorous in SF samples may, however, have limited biological degradation and the potential %BDOC could be higher than measured, thus even increasing the difference in the biodegradability of DOM between the samples of SF and those of TF and LL.

*There are no measurements/estimates/tests of soil geochemical interactions with infiltrating DOM. As indicated above, biodegradation will likely be limited in natural settings (compared to the bioincubation tests – even for bioincubation tests without the nutrient spiking). I noticed that the other reviewer also believed this to be a shortcoming of the manuscript. As gathering more data along this vein would be difficult (and is, of course, not possible for storms already past unless it was collected at the time), I recommend the other reviewers' solution: provide more discussion of geochemical controls over DOM processes within soils. Perhaps the discussion could have subsections dedicated to biological factors and geochemical factors.*

As this comment is similar to referee 1 we give the same answer.

*The meaning of DOM "origin" is unclear. For example, in the abstract: "strong significant effects of origin of ecosystem fluxes" – what is the "origin"? (A) Is it the first contact between precipitation and terrestrial surfaces (in the tree canopy), thus species specific throughfall v. stemflow v. litter leachate? Or, (B) Is it the origin of specific DOM fluorophores/molecular formulas? If (per A) the "origin" variable is used to indicate the initial DOM-enrichment process - throughfall or stemflow or litter leachate (for gap throughfall) - how is this different from the "species" variable? If the "origin" variable is used to indicate the origination of specific indicator fluorophores (like the component C1, "humic-like with terrestrial origin") or FTICR-MS formulas (like the N-rich organic compounds assumed to have atmospheric origins [p. 15, lines 10-15]), then this should be explicitly defined.*

With 'DOM origin' we mean the location where the sample was taken, thus meaning the different ecosystem fluxes in our study (species independent throughfall, stemflow, litter leachate, topsoil and subsoil solution). We will define this precisely in the manuscript.

The changes will be as follows:

Page 2, line 2: The DOM concentration and properties along the water flow path through forest ecosystems depend on its sampling location and transformation processes.

Page 2, line 10: Multivariate statistics revealed strong significant effects of ecosystem fluxes and smaller effects of main tree species on DOM quality.

Page 12, line 16: We found a significant effect of ecosystem fluxes on DOM composition variables including $SUVA_{254}$ and %PARAFAC components (PERMANOVA, p = 0.001) explaining 67 % of sample variance.

The term 'origin' associated with the description of fluorescence components and FT-ICR MS formulas is adopted from the literature and we will keep it through the manuscript.

*Although there is little literature covering throughfall and stemflow DOM quality, the authors missed some studies. Normally, one cannot cite all the studies on a particular topic; however, in this case, since so few studies exist, I recommend their inclusion. Please note that, for one of these papers, I am the lead author and it is not my intention to push my own work, only to account for the few studies on the topic. Introduction and discussion: Throughfall and stemflow DOM concentration, flux and quality (including potential sources and fates) have been reviewed and evaluated by Van Stan & Stubbins, 2018, https://doi.org/10.1002/lol2.10059. Page 16, lines 26-30: The authors only reference Qualls & Haines (1992) biodegradation estimates for throughfall. But, they do not discuss the only study reported stemflow BDOC in Qualls' recent Special Issue (Howard et al., 2018, https://doi.org/10.3390/f9050236).*

Thank you, for pointing us at these studies, which we indeed missed.

We will include information from van Stan and Stubbins (2018) as reference in the introduction section of our manuscript (Page 3 line 2)

We will include the results of Howard et al. (2018) in the discussion section (Page 17 Line 29) as follows:

This corresponded to the results of Howard et al. (2018) reporting BDOC in an interquartile range of 36-73% for cedar throughfall and stemflow samples.

Both references will be added to the reference list as followed:

Howard, D.H.; Stan, J.T.V.; Whitetree, A.; Zhu, L.; Stubbins, A. Interstorm Variability in the Biolability of Tree-Derived Dissolved Organic Matter (Tree-DOM) in Throughfall and Stemflow. Forests, 9, 236, 2018.

Van Stan, J. T. and Stubbins, A.: Tree-DOM: Dissolved organic matter in throughfall and stemflow. Limnol. Oceanogr., 3: 199-214. doi:10.1002/lol2.10059, 2018.

**Reply to Referee 3**

Dear Malak Tfaily, thank you for your detailed review of our manuscript. In the following, we hope to adequately address your constructive comments and questions.

*Page 5, line 31, did you check if 0.2 µm was enough to eliminate microbial communities originally present in the samples?*

We did not check for complete bacteria removal but filtration through 0.2 µm is a standard procedure for the removal of microorganisms. To our knowledge already a filter pore diameter of < 1 µm would be sufficient to exclude living bacterial cells so that only even smaller spores could have passed. Please note that the samples were subsequently (re-)inoculated with a microbial community extracted from the soils of the biodiversity exploratory. Therefore, the goal of the filtration was not necessarily a complete sterilization, but a standardization of the microbial community degrading the DOM in the different samples.

*Pages 4-5 what was the total number of samples and how was it distributed in terms of management? Line 16, can you give the break down for the 466 samples?*

We think, the number of analyzed samples per measurement/experiment you are asking for is given in new Table 2. Detailed information of pooled samples for FT-ICR-MS and the incubation test regarding management distribution are given in Tables S3 and S4 in the supplement. Moreover, in the supplement (Table S2), detailed numbers of samples per site, plot and ecosystem flux for the 466 fluorescence samples are given.

To better find these information, we will add the following paragraph to the material and methods section:

Page 5 line 21: An overview of sampling time and sample composition per analysis is given in Table 2. Pictures of sampling installations are given in the supporting information (Figure S1). Detailed information of selected plots per site, number of measured samples per ecosystem flux and composition of pooled samples for all measurements is provided as supporting information (Table S2: DOM characterization: fluorescence, Table S3: DOM characterization: FT-ICR-MS, Table S4: DOM biodegradability).

*Line 16, what do you mean by: To balance uneven sample numbers, we calculated mean EEMs per plot and ecosystem flux resulting in a dataset with 79 samples. Did you collapse the 466 samples into 79 samples to allow for plot versus plot comparison? what was the variability within the same plot? Line 20, how did the optical data look for these samples?*

As is visible in Table S2 the number of available sample numbers per plot and ecosystem flux was not the same for the various sample types (n>10 for TF, SF and LL; n<5 for TOP and SUB). In their tutorial review to PARAFAC, Murphy et al. (2013) caution about unequal numbers of replicated samples in PARAFAC modeling. To avoid this influence, we calculated mean EEMs to use one 'sample' per plot and ecosystem flux to gain a representative model. To give you an impression, Figure 1 shows throughfall (TF) EEM plots for the Schorfheide plot SEW8 (beech unmanaged) for different sampling dates and the resulting mean EEM used in our statistical analysis.

[Figure]

**Figure 1: Fluorescence EEMs plots of different throughfall samples of Schorfheide plot SEW8 (beech unmanaged) for different sampling dates. Mean= mean EEM calculated of all shown measurements**

*Do you believe that differences due to management is higher than that between plots within the same forest?*

Detecting differences in fluorescence spectra caused by different management practice versus intra-plot variability depends on the management categories compared. With fluorescence measurements of DOM (please keep in mind that fluorescence measurements address only the portion of DOM able to absorb and emit light) we were able to detect differences due to management decisions like tree species selection. Possible differences between differently managed forests with the same tree species, in our case unmanaged and age-class beech forests, were not statistically distinguishable.

*Page 7, line 14, only six spectra were averaged? Typically, we do at least 100.*

Our samples were measured on a FT-ICR-MS Ultra (ThermoFisher equipped with the SIMION optimized ICR cell for more homogeneous magnetic field in contrast to the first ThermoFisher edition, resulting in the specified exactness better than 1ppm). Our own working group-intern improvements led to an exactness better than 500 ppb (as example, mean deviation in this data set is 400 ppb), today. For previous studies in another institute, we used an Apex II Bruker (but of course not for DOM measurements), therefore we know some important technical differences between the ThermoFisher and the Bruker MS, which are important for the comparison of spectra from different instrument types.

The need to average large numbers of at least 100 scans is probably related to (I) the use of different mass spectrometers, namely of the suppliers Bruker and Thermo Finnigan, and (II) same term (*scan*) for different things. The Bruker machine allows to accumulate *scans* without limit, which means, the longer you measure, the more

intensive your mass peaks become (therefore, most people sum up 100-200 scans). In contrast, with Thermo instruments only an accumulation up to 50 so-called *μ-scans* is noticeable, more μ-scans do not change the signal intensity (kind of included system averaging without any possibility for changing or even excluding by the operator). Each of these 50 μ-scans is one transient, 50 μ-scans together are ~ 3 min measuring time and were combined subsequently to one so-called scan.

For further improving spectra quality, we average (no accumulation possible) the data of six such Thermo-scans (with 50 μ-scans each, in total recording time ~ 18 min).

As an example result for a beech throughfall sample we got 18010 peaks in the averaged spectrum of 6 scans, but it would be only 13003 peaks with 4 scans - all with the same intensive peaks from first scan on. (100 of such scans would imply 300 min = 5 h).

*Can you provide more details regarding formula assignment and the rules that were used? What was the number of unassigned formula? What were the ranges of C, H, N, C, O, S, P? that were used?*

Thank you for pointing out the missing ranges of formula assignments.

After first manual examinations of our sample set, we decided the following settings:

| elements | setting | comment |
|---|---|---|
| C, H, O | unlimited | |
| S, N | 0…3, | without the combinations $S_{2…3}N_3$ |
| $^{13}C$ | 0…1 | |
| P | 0 | |

- Rules (from organic mass spectrometry): for odd numbered peaks: N=0 or 2 (nitrogen rule), no $^{13}C$

- For even numbered peaks: no $^{12}CHO$ compounds; N=1 or 3 *or* $^{13}C$=1, but not in combination

- At least $O_2$ incorporated (that means, one COOH group for neg. ions), that means no pure CH compounds

- H/C ≤ 2.0

- O/C ≤ 1.4

Because of only small peaks for CHOSN, we decided to show van Krevelen plots for CHO compounds only, in contrast to other projects,

Number of formulas for beech throughfall as an example:

| | Number of formulas | comments |
|---|---|---|
| Registered peaks: | 18010 | |
| Unique identified peaks | 9878 | using the constraints above |
| Multiple identification | 2160 | These were not further considered, that means, we lose 12% of the peaks, but we exclude false positive ones |

| Not listed peaks | 5972 | most of all are $^{13}C$ isotopic peaks without further scientific information; or not assigned because not within constraints, e.g. CHOP, $CHOS_4$, $CHOS_2N_3$, $CHOS_3N_3$ …; or the peaks are so small, that their exactness is smaller than 1 ppm |
|---|---|---|

We will amend the manuscript as followed:

Page 7 Line 23: For quality control, all peaks of at least two randomly selected masses (odd and subsequent even numbered, respectively) were characterized by hand to control the exactness of the recalculated peaks and to set constraints in the calculation program as followed: C, H and O unlimited, N and S: 0–3 (without the combination $S_{>1}N_3$), $^{13}C$: 0–1 and P=0.). Molecular formulae were assigned using an in-house developed post-processing Scilab routine (Scilab Enterprises 2012).

*Figure 2, it's hard to see the zoom in but be careful about peak splitting as this can affect your formula assignment. Even though it is hard to see, it appears you had some peak splitting.*

Independent of daily tuning and calibrating, in the course of instrument life time, the peak shapes are changing (quenching of peaks after detections lead to slow adsorption of chemicals on the ICR cell walls too). For best conditions in very complex mixtures like DOM, we have got a (company developed) procedure for adjustment of ICR cell parameters (as result replacing the instrument master file). This procedure is usually not distributed to customers, because of the possibility to misalignments. The procedure is measuring defined standards the first two runs automatically (~ 1 day), followed by a manual adjustment, especially under consideration of the peak legs (should achieved < 15% peak height). That means, when the peak legs increase (independent of fresh cleaning of the ion source & quadrupole 0), we adjust the cell parameters to prevent peak splitting.

You can see randomly selected nominal masses from two different samples as an example in the following figures (Figure 2 and Figure 3). In both examples, assignments by our calculation program *and* manually are highlighted. Also impossible assignments due to multiple possibilities and/or limitation of our constraints or just because of decreasing exactness of peaks due to very small intensities are indicated. As you may see, for all peaks there is at least an explanation – and we found no signs for peak splitting or peaks without sum formula proposal.

[Figure]

Figure 2: Randomly selected nominal mass from beech throughfall sample

[Figure]

Figure 3: Randomly selected nominal mass from pine litter leachate sample

As you may see, at our Thermo instrument the base line is not visible – obviously base corrected by the manufacturer. We assume, this is the reason for missing peaks bases, which could lead to misinterpretations.

Additionally, the figures included in the discussion manuscript were converted to pdf with the whole document. For the final manuscript all figures will be submitted separately as individual files hopefully avoiding resolution losses.

**List of relevant changes**

Page 2 Line 2: The DOM concentration and properties along the water flow path through forest ecosystems depend on its sampling location and transformation processes.

Page 2 Line10: Multivariate statistics revealed strong significant effects of ecosystem fluxes and smaller effects of main tree species on DOM quality.

Page 3, line 2: The below-canopy fluxes consist of throughfall and stemflow both containing DOM of different quality (Moore et al., 2003; Inamdar et al., 2012; Levia et al., 2012; Levia and Germer, 2015; Michalzik et al., 2016; van Stan and Stubbins, 2018).

Page 3, line 27: We hypothesized *i*) that the composition and the biodegradability of DOM changes from a dominance of non-aromatic nitrogen-rich compounds of high bioavailability to highly aromatic, increasingly-oxidized, nitrogen-poor compounds with decreased bioavailability along the water flow path through forest ecosystems, from throughfall (TF), stemflow (SF), litter layer leachate (LL) to mineral topsoil (TOP) and subsoil (SUB) solution. We postulated *ii*) that aboveground changes of DOC concentrations and DOM composition are mainly controlled by selective biological degradation, while changes in mineral soil are governed by sorption to mineral surfaces. Moreover, we hypothesized *iii*) that the dominant tree species as well as forest management intensity affect the DOM composition as well as the direction and magnitude of its changes along the water flow path. The former because of the presence of species-specific compounds in DOM, like phenolic secondary metabolites in beech forests, the latter, measured as the Forest Management Index (ForMI), beside others because of its influence on the biomass production and C input into the soil (Kahl and Bauhus, 2014).

Page 4, line 4: Chemical soil properties for Hainich and Schorfheide plots are given in Table 1.

Page 5, line 21: An overview of sampling time and sample composition per analysis is given in Table 2. Pictures of sampling installations are given in the supporting information (Figure S1). Detailed information of selected plots per site, number of measured samples per ecosystem flux and composition of pooled samples for all measurements is provided as supporting information (Table S2: DOM characterization: fluorescence, Table S3: DOM characterization: FT-ICR-MS, Table S4: DOM biodegradability).

Page 7, line 23: For quality control, all peaks of at least two randomly selected masses (odd and subsequent even numbered, respectively) were characterized by hand to control the exactness of the recalculated peaks and to set constraints in the calculation program as followed: C, H and O unlimited, N and S: 0–3 (without the combination $S_{>1}N_3$), $^{13}C$: 0–1 and P=0.). Molecular formulae were assigned using an in-house developed post-processing Scilab routine (Scilab Enterprises 2012).

Page 12, line 16: We found a significant effect of ecosystem fluxes on DOM composition variables including $SUVA_{254}$ and %PARAFAC components (PERMANOVA, p = 0.001) explaining 67 % of sample variance.

Page 13, line 25: In the study of Kindler et al. (2011), the retention of DOC in mineral soil, expressed as percentage reduction of downward DOC flux, was closely related to the ratio between organic carbon content of the mineral soil and the sum of its oxalate-extractable Fe and Al content ($OC/[Fe_o+Al_o]$). This suggested that the DOC retention in mineral soils is governed by the sorption to the surfaces of Fe- and Al-(hydr)oxides. Because Fe- and Al-(hydr)oxides have a limited sorption capacity for organic matter, DOC retention in subsoils decreased

exponentially with increasing organic matter coverage of the hydroxide surfacrs (Kindler et al. 2011). In contrast to the findings of Kindler et al. (2011), we compare DOC concentrations, not fluxes. In order to test whether the data of our study fit the findings of Kindler et al. (2011), we plotted changes in DOC concentrations reported by Kindler et al. (2011) together with the data of this study against the ratio of $OC/(Fe_o+Al_o)$ in one graph (Figure 2). Different from fluxes, which always decreased with increasing soil depth in the Kindler et al. (2011) study, DOC concentrations increased with increasing depth at the Hainich sites with the highest $OC/(Fe_o+Al_o)$ ratios of all study regions (Figure 2). This increase in concentrations can be explained by a concentration effect because of evapotranspiration, in the case that the DOC sorption capacity of pedogenic Fe- and Al-(hydr)oxides is saturated. Overall, the retention of DOC in the Hainich soils of this study fitted well to the DOC retention in the European data set of Kindler et al. (2011), which showed that the regional variation of DOC retention can be as large as the variation at continental scale. The reduction of DOC concentrations between TOP and SUB significantly decreased with increasing $OC/(Fe_o+Al_o)$ ratio (p = 0.027; Figure 2), corroborating the hypothesis that sorption to pedogenic Fe- and Al-(hydr)oxides controled DOC retention in mineral soils (Kindler et al. 2011). However, the results for mineral soils of the Schorfheide sites did not follow this pattern, as DOC concentrations decreased from TOP to SUB by 33–72% regardless of the $OC/(Fe_o+Al_o)$ ratio (Figure 2). At the Schorfheide sites, other processes than sorption to Fe- and Al-(hydr)oxide surfaces likely governed DOC retention. The Schorfheide soils developed from fluvioglacial quartzitic sands covering carbonate-free glacial till. Because of their poor pH buffering capacity, these soils were very acidic ($pH_{CaCl2}$ = 3.0–3.6 in topsoils). The mean pH value in soil water samples of the Schorfheide sites was 4.5 in TOP solutions, increasing to 5.5 in SUB solutions. This means that Al-(hydr)oxides were dissolved in the Schorfheide topsoils, increasing $Al^{3+}$-concentrations in soil water and leachates. The pH increase to 5.5 along the way from TOP to SUB likely induced a re-precipitation of Al. We assume that dissolved organic matter transported from TOP to SUB co-precipitated together with $Al^{3+}$ as described by Nierop et al. (2002) and Jansen et al. (2003, 2005) for acidic sandy soils from the Netherlands. If DOM was immobilized as insoluble metal-organic matter precipitate in B-horizons, no limitation by available sorption sites of surfaces of pedogenic (hydr)oxides would apply, so that reductions of DOC concentrations with increasing depth in mineral soil would be independent of the soils $OC/(Fe_o+Al_o)$ saturation index.

Page 17, line 29: This corresponded to the results of Howard et al. (2018) reporting BDOC in an interquartile range of 36-73% for cedar throughfall and stemflow samples.

Page 17, line 32: Besides other factors, nutrient availability can affect biological degradation of organic matter in ecosystem samples. In our study no additional nutrients were added to compensate for possible limitations. We calculated maximum nutrient demands for the mineralized organic carbon in our samples by using values for bacterial growth requirement of nitrogen and phosphorus suggested by Felmann et al. (2008) and measured concentrations of N and P in the solution samples prior to pooling for the incubation experiment (Table S5). The results suggested that constrained biodegradation due to nutrient limitation in TF and LL samples was not likely. Low concentrations for phosphorus in SF samples may, however, have had a limiting effect of biological degradation and the amount of %BDOC could be higher than measured, thus even increasing the difference in the biodegradability of DOM between the samples of SF and those of TF and LL.

Page 19, line 17: Ad-Hoc-Arbeitsgruppe Boden der Staatlichen Geologischen Dienste und der Bundesanstalt für Geowissenschaften und Rohstoffe: Bodenkundliche Kartieranleitung (KA5), Schweitzerbart´sche Verlagsbuchhandlung, Stuttgart, 141–142, 2005.

Page 21, line 17: Fischer, M., Bossdorf, O., Gockel, S., Hänsel, F., Hemp, A., Hessenmöller, D., Korte, G., Nieschulze, J., Pfeiffer, S., Prati, D., Renner, S., Schöning, I., Schumacher, U., Wells, K., Buscot, F., Kalko, E. K.V., Linsenmair, K.E., Schulze, E-D., Weisser, W. W.: Implementing large-scale and long-term functional biodiversity research: The Biodiversity Exploratories, Basic and Applied Ecology, 11, 473-485, doi.org/10.1016/j.baae.2010.07.009, 2010.

Page 22, line 13: Howard, D.H.; Stan, J.T.V.; Whitetree, A.; Zhu, L.; Stubbins, A. Interstorm Variability in the Biolability of Tree-Derived Dissolved Organic Matter (Tree-DOM) in Throughfall and Stemflow. Forests, 9, 236, 2018.

Page 22, line 21: Jansen, B., Nierop, K.G.J., and Verstraten, J.M.: Mobility of Fe(II), Fe(III) and Al in acidic forest soils mediated by dissolved organic matter: influence of solution pH and metal/organic carbon ratios. Geoderma, 113, 323– 340, 2003.

Page 22, line 23: Jansen, B., Nierop, K.G.J., and Verstraten, J.M.: Mechanisms controlling themobility of dissolved organic matter, aluminium and iron in podzol B horizons. Eur. J. Soil Sci., 56, 537–550. doi: 10.1111/j.1365-2389.2004.00686.x, 2005.

Page 24, line 29: Nierop, K.G.J., Jansen, B., and Verstraten, J.M.: Dissolved organic matter, aluminium and iron interactions: precipitation induced by metal/carbon ratio, pH and competition. Sci. Total. Environ., 300, 201–211, 2002.

Page 26, line 25: Van Stan, J. T. and Stubbins, A.: Tree-DOM: Dissolved organic matter in throughfall and stemflow. Limnol. Oceanogr., 3: 199-214. doi:10.1002/lol2.10059, 2018.

A new Table 1 was added to the manuscript. All Table numbers shifted accordingly.

Table S6 was added to the supplement.

A new Figure 2 was added to the manuscript. All Figure numbers shifted accordingly.

**Marked-up manuscript version**

[revised manuscript text omitted]
 *i*) that the composition and the biodegradability of DOM changes from a dominance of non-aromatic nitrogen-rich compounds of high bioavailability to highly aromatic, increasingly-oxidized, nitrogen-poor compounds with decreased bioavailability along the water flow path through forest ecosystems, from throughfall (TF), stemflow (SF), litter layer leachate (LL) to mineral topsoil (TOP) and subsoil (SUB) solution. We postulated *ii*) that aboveground changes of DOC concentrations and DOM composition are mainly controlled by selective biological degradation, while changes in mineral soil are governed by sorption to mineral surfaces. Moreover, we hypothesized *iii*) that the dominant tree species as well as forest management intensity affect the DOM composition as well as the direction and magnitude of its changes along the water flow path. The former because of the presence of species-specific compounds in DOM, like phenolic secondary metabolites in beech forests, the latter, measured as the Forest Management Index (ForMI), beside others because of its influence on the biomass production and C input into the soil (
[revised manuscript text omitted]

In the study of Kindler et al. (2011), the retention of DOC in mineral soil, expressed as percentage reduction of downward DOC flux, was closely related to the ratio between organic carbon content of the mineral soil and the sum of its oxalate-extractable Fe and Al content (OC/[$Fe_o$+$Al_o$]). This suggested that the DOC retention in mineral soils is governed by the sorption to the surfaces of Fe- and Al-(hydr)oxides. Because Fe- and Al-(hydr)oxides have a limited sorption capacity for organic matter, DOC retention in subsoils decreased exponentially with increasing organic matter coverage of the hydroxide surfacrs (Kindler et al. 2011). In contrast to the findings of Kindler et al. (2011), we compare DOC concentrations, not fluxes. In order to test whether the data of our study fit the findings of Kindler et al. (2011), we plotted changes in DOC concentrations reported by Kindler et al. (2011) together with

the data of this study against the ratio of $OC/(Fe_o+Al_o)$ in one graph (Figure 2). Different from fluxes, which always decreased with increasing soil depth in the Kindler et al. (2011) study, DOC concentrations increased with increasing depth at the Hainich sites with the highest $OC/(Fe_o+Al_o)$ ratios of all study regions (Figure 2). This increase in concentrations can be explained by a concentration effect because of evapotranspiration, in the case that the DOC sorption capacity of pedogenic Fe- and Al-(hydr)oxides is saturated. Overall, the retention of DOC in the Hainich soils of this study fitted well to the DOC retention in the European data set of Kindler et al. (2011), which showed that the regional variation of DOC retention can be as large as the variation at continental scale. The reduction of DOC concentrations between TOP and SUB significantly decreased with increasing $OC/(Fe_o+Al_o)$ ratio ($p = 0.027$; Figure 2), corroborating the hypothesis that sorption to pedogenic Fe- and Al-(hydr)oxides controled DOC retention in mineral soils (Kindler et al. 2011). However, the results for mineral soils of the Schorfheide sites did not follow this pattern, as DOC concentrations decreased from TOP to SUB by 33–72% regardless of the $OC/(Fe_o+Al_o)$ ratio (Figure 2). At the Schorfheide sites, other processes than sorption to Fe- and Al-(hydr)oxide surfaces likely governed DOC retention. The Schorfheide soils developed from fluvioglacial quartzitic sands covering carbonate-free glacial till. Because of their poor pH buffering capacity, these soils were very acidic ($pH_{CaCl2}$ = 3.0–3.6 in topsoils). The mean pH value in soil water samples of the Schorfheide sites was 4.5 in TOP solutions, increasing to 5.5 in SUB solutions. This means that Al-(hydr)oxides were dissolved in the Schorfheide topsoils, increasing $Al^{3+}$-concentrations in soil water and leachates. The pH increase to 5.5 along the way from TOP to SUB likely induced a re-precipitation of Al. We assume that dissolved organic matter transported from TOP to SUB co-precipitated together with $Al^{3+}$ as described by Nierop et al. (2002) and Jansen et al. (2003, 2005) for acidic sandy soils from the Netherlands. If DOM was immobilized as insoluble metal-organic matter precipitate in B-horizons, no limitation by available sorption sites of surfaces of pedogenic (hydr)oxides would apply, so that reductions of DOC concentrations with increasing depth in mineral soil would be independent of the soils $OC/(Fe_o+Al_o)$ saturation index.

[revised manuscript text omitted]

**Acknowledgements**

We thank the managers of the three Exploratories, Swen Renner, Kirsten Reichel-Jung, Sonja Gockel, Kerstin Wiesner, Katrin Lorenzen, Andreas Hemp, Martin Gorke, and all former managers for their work in maintaining the plot and project infrastructure; Simone Pfeiffer, Maren Gleisberg, and Christiane Fischer for giving support through the central office, Jens Nieschulze, and Michael Owonibi for managing the central data base, and Eduard Linsenmair, Dominik Hessenmöller, Daniel Prati, François Buscot, Ernst-Detlef Schulze, Wolfgang W. Weisser, and the late Elisabeth Kalko for their roles in setting up the Biodiversity Exploratories project We gratefully acknowledge the support of Corinna Voss, Minh-Chi Tran-Thi and Robert Jonov during processing and analysis of samples. We thank Marion Schrumpf, Ingo Schöning, Nadine Herold and Theresa Klötzing for providing soil chemical data. Furthermore, we thankMalak Tfaily, John van Stan and one anonymous referee, whose constructive comments helped to improve this manuscript.

The work has been funded by the DFG Priority Program 1374 "Infrastructure Biodiversity Exploratories" (contributing project BECycles, Wi 1601/12, Ka 1139/17, Mi 927/2, Si 1106/4). Field work permits were issued by the responsible state environmental offices of Baden-Württemberg, Thüringen, and Brandenburg (according to §72 BbgNatSchG).

**References**

Ad-Hoc-Arbeitsgruppe Boden der Staatlichen Geologischen Dienste und der Bundesanstalt für Geowissenschaften und Rohstoffe: Bodenkundliche Kartieranleitung (KA5), Schweizerbart´sche Verlagsbuchhandlung, Stuttgart, 141–142, 2005.

[revised manuscript text omitted]

**Tables**

Table 2: Chemical soil properties and mean dissolved organic carbon (DOC) concentrations of plots in the Hainich Dün (HEW) and Schorfheide Chorin (SEW) sites. LL = litter leachate, TOP = topsoil, SUB = subsoil, reduction cDOC (%) = reduction of DOC concentration in % between LL and TOP or TOP and SUB, Corg = organic carbon content of soil, $Al_0$ = aluminum content extracted with ammonium oxalate, $Fe_0$ = iron content extracted with ammonium oxalate

| plot | ecosystem flux / soil layer | management category | DOC [mg/L] | reduction cDOC (%) | $C_{org}$ [g/kg] | $Al_o$ [g/kg] | $Fe_o$ [g/kg] | clay [g/kg] | texture (KA5*) | pH soil (CaCl₂) |
|------|------|------|------|------|------|------|------|------|------|------|
| HEW1 | LL | category | 39.23 | | | | | | | |
| HEW1 | Top | coniferous age-class | 11.26 | 71.31 | 69.14 | 3.28 | 3.50 | 326 | Lu | 7.0 |
| HEW1 | Sub | coniferous age-class | 15.46 | -37.36 | 28.99 | 4.38 | 3.89 | 239 | Uls/Tl | 7.5 |
| HEW2 | LL | coniferous age-class | 41.54 | | | | | | | |
| HEW2 | Top | coniferous age-class | 24.72 | 40.49 | 50.60 | 1.43 | 4.92 | 241 | Lu /Ut4 | 4.6 |
| HEW2 | Sub | coniferous age-class | 7.70 | 68.84 | 6.95 | 1.73 | 2.98 | 589 | Tu2 | 7.0 |
| HEW3 | LL | coniferous age-class | 66.08 | | | | | | | |
| HEW3 | Top | coniferous age-class | 16.76 | 74.64 | 47.74 | 2.33 | 3.18 | 359 | Ut3/Ut2 | 3.9 |
| HEW3 | Sub | coniferous age-class | 14.04 | 16.22 | 10.33 | 2.38 | 2.37 | 634 | Tl | 6.7 |
| HEW4 | LL | deciduous age-class | 22.76 | | | | | | | |
| HEW5 | LL | deciduous age-class | 18.26 | | | | | | | |
| HEW5 | Top | deciduous age-class | 7.50 | 58.92 | 61.77 | 3.79 | 3.19 | 457 | Lu | 5.2 |
| HEW5 | Sub | deciduous age-class | 5.12 | 31.78 | | | | | | 7.2 |
| HEW6 | LL | deciduous age-class | 17.57 | | | | | | | |
| HEW6 | Top | deciduous age-class | 11.30 | 35.70 | 34.40 | 2.19 | 3.73 | 214 | Lu | 4.3 |
| HEW6 | Sub | deciduous age-class | 5.20 | 54.02 | 5.15 | 2.45 | 3.62 | 442 | Tu2/Tl | 5.4 |
| HEW10 | LL | unmanaged | 24.18 | | | | | | | |
| HEW10 | Top | unmanaged | 7.95 | 67.15 | 67.59 | 3.49 | 4.74 | 485 | Ut4 | 4.1 |
| HEW11 | LL | unmanaged | 29.77 | | | | | | | |
| HEW11 | Top | unmanaged | 10.96 | 63.20 | 58.52 | 3.31 | 4.72 | 404 | Ut4 | 4.9 |
| HEW11 | Sub | unmanaged | 12.10 | -10.41 | 19.78 | 3.46 | 4.32 | 517 | Tu3 | 4.9 |
| HEW12 | LL | unmanaged | 24.02 | | | | | | | |
| HEW12 | Top | unmanaged | 7.42 | 69.09 | 31.13 | 1.72 | 2.64 | 164 | Ut4 | 3.9 |
| HEW12 | Sub | unmanaged | 5.60 | 24.52 | 5.58 | 2.43 | 3.19 | 424 | Tu2 | 5.9 |
| SEW1 | LL | coniferous age-class | 67.07 | | | | | | | |
| SEW1 | Top | coniferous age-class | 58.63 | 12.59 | 18.34 | 1.82 | 2.02 | 5 | Sl2 | 3.6 |
| SEW1 | Sub | coniferous age-class | 16.19 | 72.39 | 2.06 | 2.05 | 2.03 | 1 | Sl2 | 3.9 |
| SEW2 | LL | coniferous age-class | 58.50 | | | | | | | |
| SEW2 | TOP | coniferous age-class | 26.73 | 54.31 | 16.99 | 1.78 | 1.94 | 32 | Sl2 | 3.5 |
| SEW2 | Sub | coniferous age-class | 11.40 | 57.34 | 2.26 | 2.68 | 2.50 | 33 | Sl2 | 4.2 |
| SEW3 | LL | coniferous age-class | 57.20 | | | | | | | |
| SEW3 | Top | coniferous age-class | 37.09 | 35.15 | 20.95 | 1.61 | 1.62 | 17 | Sl2 | 3.3 |
| SEW3 | Sub | coniferous age-class | 15.06 | 59.39 | 4.05 | 2.09 | 1.38 | 3 | Sl2 | 4.0 |
| SEW5 | LL | deciduous age-class | 32.84 | | | | | | | |
| SEW5 | Top | deciduous age-class | 91.81 | -179.59 | 29.56 | 1.20 | 1.04 | 1 | Sl2 | 3.1 |
| SEW5 | Sub | deciduous age-class | 27.86 | 69.65 | 2.50 | 2.21 | 1.29 | 1 | Sl2/Su2 | 3.4 |
| SEW6 | LL | deciduous age-class | 37.84 | | | | | | | |
| SEW6 | Top | deciduous age-class | 12.84 | 66.05 | 31.05 | 2.39 | 2.52 | 23 | Sl2 | 3.4 |
| SEW6 | Sub | deciduous age-class | 8.48 | 34.00 | 1.45 | 1.77 | 1.60 | 17 | Sl2 | 3.9 |
| SEW7 | LL | unmanaged | 26.20 | | | | | | | |
| SEW7 | Top | unmanaged | 46.86 | -78.84 | 24.30 | 1.74 | 1.78 | 1 | Sl2 | 3.2 |
| SEW7 | Sub | unmanaged | 16.89 | 63.96 | 6.37 | 1.38 | 1.55 | | Sl2 | 3.7 |
| SEW8 | LL | unmanaged | 41.33 | | | | | | | |
| SEW8 | Top | unmanaged | 29.03 | 29.76 | 29.20 | 1.86 | 1.58 | 20 | Sl2 | 3.1 |
| SEW8 | Sub | unmanaged | 13.07 | 54.97 | 10.28 | 1.52 | 1.48 | 1 | Sl2 | 3.2 |
| SEW9 | LL | unmanaged | 42.50 | | | | | | | |
| SEW9 | Top | unmanaged | 39.94 | 6.01 | 22.96 | 0.95 | 1.01 | 18 | Sl2 | 3.0 |
| SEW9 | Sub | unmanaged | 14.92 | 62.65 | 4.81 | 1.43 | 1.09 | 1 | Sl2 | 3.7 |

* KA5 = Ad-Hoc-Arbeitsgruppe Boden (2005)

[revised manuscript text omitted]